# Reflection-Window Decoding: Text Generation with Selective Refinement

**Zeyu Tang** [* 1]  **Zhenhao Chen** [* 2]  **Xiangchen Song** [1]  **Loka Li** [2]  **Yunlong Deng** [2]  **Yifan Shen** [2]  **Guangyi Chen** [1 2]
**Peter Spirtes** [1]  **Kun Zhang** [1 2]

## Abstract

The autoregressive decoding for text generation in large language models (LLMs), while widely used, is inherently suboptimal due to the lack of a built-in mechanism to perform refinement and/or correction of the generated content. In this paper, we consider optimality in terms of the joint probability over the generated response, when jointly considering all tokens at the same time. We theoretically characterize the potential deviation of the autoregressively generated response from its globally optimal counterpart that is of the same length. Our analysis suggests that we need to be cautious when noticeable uncertainty arises during text generation, which may signal the sub-optimality of the generation history. To address the pitfall of autoregressive decoding for text generation, we propose an approach that incorporates a sliding reflection window and a pausing criterion, such that refinement and generation can be carried out interchangeably as the decoding proceeds. Our selective refinement framework strikes a balance between efficiency and optimality, and our extensive experimental results demonstrate the effectiveness of our approach.

## 1. Introduction

It is a well-known insight in optimization theory that coordinate-wise optimization, conditioned on the previously optimized arguments along each axis sequentially, generally does not guarantee finding the global optimum (Törn & Žilinskas, 1989; Nocedal & Wright, 1999). Similarly, in the context of decoding in large language models (LLMs), expecting to achieve a globally optimal response by sequentially accumulating per-token optimal decisions, as done in

---
[*]Equal contribution [1]Carnegie Mellon University [2]Mohamed bin Zayed University of Artificial Intelligence. Correspondence to: Zeyu Tang <zeyutang@cmu.edu>, Zhenhao Chen <zhenhao.chen@mbzuai.ac.ae>.

*Proceedings of the $42^{nd}$ International Conference on Machine Learning*, Vancouver, Canada. PMLR 267, 2025. Copyright 2025 by the author(s).

purely autoregressive decoding, may be overly optimistic. Despite significant recent progress (Vaswani et al., 2017; Radford et al., 2019; Brown et al., 2020; OpenAI, 2023; Touvron et al., 2023; Gemini Team, 2023; Llama Team, 2024; Gemma Team, 2024; Abdin et al., 2024a; DeepSeek-AI, 2025), how to approach the optimal text that one can possibly sample from a model still remains an open question.

Previous works have demonstrated challenges faced by autoregressive decoding in terms of the handling of long sequences (Wu et al., 2021) and the inefficient inference (Lin et al., 2021; Li et al., 2022). At the level of decoding, other than the traditional greedy search, Holtzman et al. (2020) proposed Top-$p$ sampling (also known as Nucleus Sampling), a stochastic method that adjusts the next-token set based on the shape of conditional probability mass function. Alternatively, different from the cumulated probability mass, Top-$k$ sampling limits the number of available options when sampling for the next token (Fan et al., 2018; Holtzman et al., 2018; Radford et al., 2019). Another empirical technique involves adjusting the shape of probability distribution with the temperature hyperparameter (Fan et al., 2018; Holtzman et al., 2018; Shih et al., 2023; Peeperkorn et al., 2024), or decoding as optimization with more than one models (Li et al., 2023).

In order to improve the sampling efficiency, speculative decoding approaches have been proposed, where multiple tokens are predicted in parallel as if one were sampling from the model (or its lighter counterpart) repetitively (Leviathan et al., 2023; Chen et al., 2023a; Xia et al., 2023; Kim et al., 2024; Sun et al., 2024). Efficient inference with beam search (Xie et al., 2024; Zhu et al., 2024; Wei et al., 2024; Yang et al., 2024b) and probabilistic programs (Lew et al., 2023) have been explored in the recent literature. Improving generated content through high-level behaviors, e.g., through instructing self-correction or conducting self-improvement with external or internal feedback (Yao et al., 2022; Bai et al., 2022; Pan et al., 2023; Shinn et al., 2023; Ganguli et al., 2023; Chen et al., 2023b; Kim et al., 2023; Tyen et al., 2023; Madaan et al., 2024), has also been studied.

While previous literature has explored various methods to enhance generated content, the fundamental limitation of autoregressive decoding for text generation remains

under-explored. This gap represents a distinct perspective, different from high-level model behaviors (e.g., self-refinement studied in Madaan et al. 2024) or inference efficiency (e.g., various speculative decoding methods surveyed in Xia et al. 2024). In this paper, we theoretically characterize the inherent shortcoming of autoregressive decoding and propose an empirical method to mitigate this issue. Our contributions can be summarized as follows:

- We theoretically characterize the sub-optimality of autoregressive decoding for text generation, demonstrating its lack of a built-in mechanism to perform refinement/correction of generated content at the decoding level.

- We propose a framework involving a sliding reflection window and a corresponding pause criterion, enabling an interchangeable process of refinement and generation.

- Through extensive empirical evaluations, our approach demonstrates significant improvement over existing decoding approaches, and maintains performance comparable or superior to beam search while being more efficient.

## 2. Motivation and High-Level Illustration

In this section, we first present our motivations behind addressing the inherent shortcoming of purely autoregressive decoding for text generation (Section 2.1). Then in Section 2.2, we present a high-level summary of our proposed approach involving interchangeably switching between refinement (upon reflection on previously generated content) and generation (of the additional new content).

### 2.1. Inherent Shortcoming of Autoregressive Decoding

Insight from the optimization theory highlights the gap between coordinate-wise accumulation of optimum and the global optimum (Törn & Žilinskas, 1989; Nocedal & Wright, 1999). Recent research advances in cognitive linguistics have also argued that language is primarily a tool for communication (for humans) rather than thought (Fedorenko et al., 2024). While language naturally unfolds in a one-dimensional sequence, its underlying dependence pattern extends beyond a purely autoregressive structure.

Let us consider an example of writing a novel. For a long-format writing like novels, outlining (also referred to as plotting) is essential for structuring ideas, planning narratives, and crafting engaging drafts (King, 2000; Serravallo, 2017). Sub-goals refer to relatively small and achievable tasks that guide the author through each stage of the story, for instance, the setting of the circumstance, the element of tension and emotion, the sensory imagination of the scene.

As we illustrate in Figure 1(a), $X_i^*$'s represent words or phrases in the novel, and $S_i$'s represent sub-goals, which may be related in a hierarchical way. For instance,

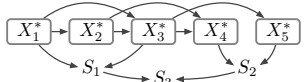 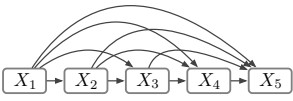

(a) (Sub-)goals jointly satisfied by the optimal sequence

(b) Purely autoregressive way of decoding for text generation

Figure 1: Illustrative diagrams of different dependence patterns among variables representing tokens or phrases in text generation. Panel (a): the dependence pattern among variables in the optimal sequence where there are (sub-)goals $S_i$'s to achieve, specifying conditions or constraints that should *jointly* be satisfied by $X_i^*$'s. Panel (b): the autoregressive way of text generation, where $X_i$ is only allowed to depend on $X_j$ if $j < i$.

sub-goals within a single scene altogether serve the purpose of furthering the development of the story. We model sub-goals in terms of selection variables $S_i$ since they represent constraints or objectives to achieve, which involve certain criteria to be satisfied over the variables that they operate upon.[1] As we can see from Figure 1(a), the variables in optimal sequence (the novel in this example) $X_i^*$'s *jointly* satisfy criteria, or optimize objectives, specified by sub-goals $S_i$'s. This indicates that the best $X_i^*$ in the optimal sequence depends on best values of all other $X_j^*$'s. However, with an autoregressive way of text generation, as illustrated in Figure 1(b), we only allow $X_i$ to depend on $X_j$'s if $j < i$, which is clearly sub-optimal.

### 2.2. Selective Refinement Through Reflection Window

As illustrated in Section 2.1, one outcome of limiting the dependence pattern to the autoregressive structure is the lack of a built-in mechanism to correct or refine previously generated content at the decoding level, since what was generated is not subject to further edit (if without further processing). In this subsection, we present a high-level summary of our approach that attempts to address this issue.[2] Specifically, we propose a selective refinement framework that facilitates an interchangeable process of refinement and generation, so that the overall response satisfies requirements or objectives that operate jointly over all involved tokens.

As illustrated in Figures 2 and 3, we can use fast and slow pointers on the generated content to form segments of a certain length, namely, the sliding reflection window, and perform potential refinements within this sliding window as the text generation proceeds.[3] Our reflection-window decoding framework allows for revision before the entire output is completed, which offers several advantages.

---

[1]This modeling choice is consistent with the modeling of causal relations among variables of interest through selection variables in a directed acyclic graph (DAG) (Spirtes et al., 1993; Pearl, 2009).

[2]We present in detail our technical approach in Section 5.1.

[3]The naming of pointers is motivated by Kahneman (2011).

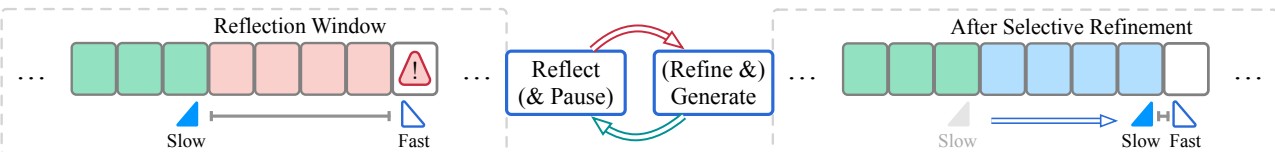

Figure 2: Overview of our approach to address the inherent shortcoming of autoregressive decoding for text generation, where the LLM interchangeably switches between refinement and generation. The fast pointer pauses if the *pausing criterion* is triggered, indicating the existence of a potential issue in the generated history. Then, the model *refines/corrects* the content between the fast and slow pointers before continuing generation, so that the slow pointer catches up with the fast pointer before the latter can move on. Reflection window refers to the content between the fast and slow pointers.

First, we can improve the generated content in a timely manner. If there are multiple potential issues in the generation history, the revision after finishing the generation can be inefficient since we allow errors to accumulate. In other words, without a built-in mechanism for refinement or correction at the decoding level, we are forced to rely on high-level model behaviors and operate at a coarser granularity. This often involves regenerating entire sentences (rather than refining words or phrases), and/or editing through multiple iterations (rather than interchangeably generate and refine in a single run), as in self-correction and self-improvement approaches (Yao et al., 2022; Bai et al., 2022; Pan et al., 2023; Shinn et al., 2023; Ganguli et al., 2023; Chen et al., 2023b; Kim et al., 2023; Tyen et al., 2023; Madaan et al., 2024).

Second, our focus on selective refinement during decoding is not solely driven by inference efficiency considerations. The primary goal of previous approaches, e.g., speculative decoding (Leviathan et al., 2023; Chen et al., 2023a; Xia et al., 2023; Kim et al., 2024; Sun et al., 2024; Xia et al., 2024), is to accelerate sampling from (a lighter version of) the original model, while the underlying decoding mechanism remains purely autoregressive.

Third, due to the one-dimensional progression of text generation, our sliding reflection window mechanism, given a pausing criterion, enables timely and assured detection of issues in the generated text. Our framework complements previous approaches, and furthermore, offers versatility. One can incorporate *pausing criteria* and *refinement/correction methods* at the decoding level, while preserving the ability to further leverage strategies that rely on high-level behaviors.

The empirical pausing criteria we use (detailed in Section 5) are guided by our theoretical characterization of the sub-optimality of autoregressive text generation, and to this theoretical analysis we now turn.

## 3. Theoretical Characterization of the Sub-Optimality of Autoregressive Decoding

We theoretically characterize the sub-optimality of autoregressive text generation. We show that even if an LLM is sufficiently trained and can perfectly capture any autoregressive decomposition of the joint density, the autoregressive way of text generation can still deviate from the globally optimal response, even for the well-defined objective of maximizing output probability given a fixed length (setting aside whether this objective fully aligns with the ultimate goal).

Let us denote a token from the vocabulary $\mathcal{V}$ as $w_v \in \mathcal{V}$, whose index in the vocabulary is $v \in |\mathcal{V}|$. We use "$i : j$" to denote the increasing integer sequence from $i$ to $j$ if $i \leq j$, e.g., $1 : t \coloneqq 1, 2, \ldots, t$ if $t > 1$, otherwise, $i : j \coloneqq \varnothing$.

**Definition 3.1** (Globally Optimal Length-$T$ Response). We say a sequence $w_{\mathbf{v}_T^*[1]} w_{\mathbf{v}_T^*[2]} \ldots w_{\mathbf{v}_T^*[T]}$ is globally optimal among all possible length-$T$ responses following the prompt $X_{\leq 0}$, if it has the highest ground-truth conditional probability, denoted by $f(X_{1:t} \mid X_{\leq 0})$ where $t \in [1, T]$:

$$\begin{aligned} \mathbf{v}_T^* &= (\mathbf{v}_T^*[1], \mathbf{v}_T^*[2], \ldots, \mathbf{v}_T^*[T]) \\ &\coloneqq \operatorname*{argmax}_{v_i \in |\mathcal{V}|, i=1,2,\ldots,T} f(X_{1:T} = w_{v_1} w_{v_2} \ldots w_{v_T} \mid X_{\leq 0}). \end{aligned} \quad (1)$$

**Definition 3.2** (Stepwise Optimal Length-$T$ Response). We say a sequence $w_{\widehat{\mathbf{v}}_T[1]} w_{\widehat{\mathbf{v}}_T[2]} \ldots w_{\widehat{\mathbf{v}}_T[T]}$ is stepwise optimal for prompt $X_{\leq 0}$, if the sequence consists of tokens that correspond to highest token-by-token conditional probabilities, denoted by $g(X_t \mid X_{1:t-1}, X_{\leq 0})$ where $t \in [1, T]$:[4]

$$\begin{aligned} \widehat{\mathbf{v}}_T[1] &\coloneqq \operatorname*{argmax}_{v_1 \in |\mathcal{V}|} g(X_1 = w_{v_1} \mid X_{\leq 0}), \\ \widehat{\mathbf{v}}_T[2] &\coloneqq \operatorname*{argmax}_{v_2 \in |\mathcal{V}|} g(X_2 = w_{v_2} \mid X_1 = w_{\widehat{\mathbf{v}}_T[1]}, X_{\leq 0}), \\ &\cdots \qquad\qquad\qquad\qquad\qquad\qquad\qquad (2) \\ \widehat{\mathbf{v}}_T[T] &\coloneqq \operatorname*{argmax}_{v_T \in |\mathcal{V}|} g\left(\begin{array}{l} X_T = w_{v_T} \mid X_{\leq 0}, \text{ and} \\ X_{1:T-1} = w_{\widehat{\mathbf{v}}_T[1]} \ldots w_{\widehat{\mathbf{v}}_T[T-1]} \end{array}\right), \end{aligned}$$

and $\widehat{\mathbf{v}}_T = (\widehat{\mathbf{v}}_T[1], \widehat{\mathbf{v}}_T[2], \ldots, \widehat{\mathbf{v}}_T[T])$.

In general, the longer the sequence, the lower the joint probability tends to be. The fair comparison of optimality should be length-specific, and the optimal response of a shorter

---

[4]For notional clarity, instead of $g(X_t \mid X_{1:t-1} = \mathbf{x}_{1:t-1}, X_{\leq 0} = \mathbf{x}_{\leq 0})$, we use shorthand notations in the conditioning set, i.e., $g(X_t \mid X_{1:t-1}, X_{\leq 0})$. We will adopt this simplification throughout the paper, as long as it remains unambiguous.

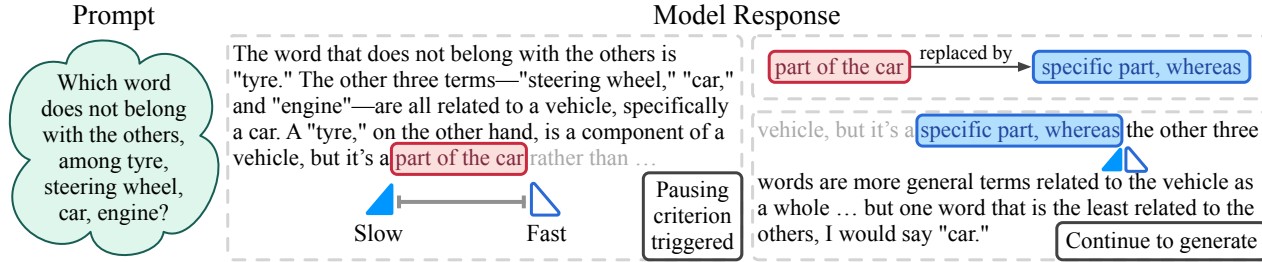

Figure 3: A concrete example demonstrating our reflection-window decoding.

length is not necessarily identical to the prefix of the optimal response that is longer in length. For instance, if we were to use 10 words to distinguish between joint and conditional densities, one might say "*joint density combines all variables; conditional adjusts for known variables.*" However, if we can use 15 words, one might say "*joint density reflects combined probabilities of all variables; conditional density adjusts probabilities given known variables.*" Therefore, we explicitly keep the length $T$ in the notation of vocabulary indices of tokens that constitute the length-$T$ responses.

**Assumption 3.3** (Oracle LLM). We say an autoregressive LLM is an oracle LLM, if the following relation holds for any response of a length $T \geq 1$:

$$f(X_{1:T} \mid X_{\leq 0}) = \Pi_{t=1}^{T} g(X_t \mid X_{1:t-1}, X_{\leq 0}). \quad (3)$$

Assumption 3.3 specifies that, after given the prompt or generated text history $X_{\leq 0}$, an oracle (or very well-trained) LLM can recover the ground-truth probability of $X_{1:T}$ as a whole follows $X_{\leq 0}$, by multiplying token-by-token generating probabilities in an autoregressive way.[5] We would like to note that Assumption 3.3 only states that an oracle LLM can perfectly capture the autoregressive way of probability partitioning of text sequences, and this itself does *not* guarantee the equivalence between stepwise optimal response and the same-length globally optimal response for $T > 1$.[6]

**Assumption 3.4** (Strict Preference Among Same-Length Sequences). For any two length-$T$ different sequences following the prompt $X_{\leq 0}$, there is a strict preference between them in terms of the ground-truth conditional probability $f(X_{1:T} \mid X_{\leq 0})$. In other words, the ground-truth conditional probabilities of two length-$T$ sequences equal to each other if and only if the sequences are identical.

Assumption 3.4 specifies that from the ground-truth conditional probability perspective, there is a strict preference between how well two different same-length responses fol-

low the prompt $X_{\leq 0}$, i.e., the ground-truth probability mass function $f(X_{1:T} \mid X_{\leq 0})$ is injective for any given $T > 0$.

**Assumption 3.5** (Irreversible Advantage Once Manifested). When a stepwise optimal length-$T$ response from an oracle (Assumption 3.3) autoregressive LLM $w_{\widehat{\mathbf{v}}_T[1]} w_{\widehat{\mathbf{v}}_T[2]} \ldots w_{\widehat{\mathbf{v}}_T[T]}$ is not the globally optimal length-$T$ response $w_{\mathbf{v}_T^*[1]} w_{\mathbf{v}_T^*[2]} \ldots w_{\mathbf{v}_T^*[T]}$, then if the deviation manifests itself at the length-$L$ $(1 < L \leq T)$ prefix-sequences, the advantage of the globally optimal length-$T$ response will not be reversed afterwards:

$$\text{if } \exists L \in (1, T], f(X_{1:L} = w_{\widehat{\mathbf{v}}_T[1]} w_{\widehat{\mathbf{v}}_T[2]} \ldots w_{\widehat{\mathbf{v}}_T[L]} \mid X_{\leq 0})$$
$$< f(X_{1:L} = w_{\mathbf{v}_T^*[1]} w_{\mathbf{v}_T^*[2]} \ldots w_{\mathbf{v}_T^*[L]} \mid X_{\leq 0}),$$
$$\text{then } \forall M \in [L, T], f(X_{1:M} = w_{\widehat{\mathbf{v}}_T[1]} \ldots w_{\widehat{\mathbf{v}}_T[M]} \mid X_{\leq 0})$$
$$< f(X_{1:M} = w_{\mathbf{v}_T^*[1]} \ldots w_{\mathbf{v}_T^*[M]} \mid X_{\leq 0}).$$

Assumption 3.5 specifies that if the advantage (in terms of a higher ground-truth conditional probability) of the globally optimal length-$T$ sequence can be observed at the length-$L$ prefix-sequence, such advantage will not be reversed when considering longer prefix-sequences.

**Theorem 3.6** (Indication of Deviation from the Globally Optimal Length-$T$ Response). *Given the prompt $X_{\leq 0}$, when an oracle LLM (Assumption 3.3) generates a stepwise optimal length-$T$ response which is not the globally optimal response with the same length, let $L \leq T$ denote the minimum length of prefix-sequence needed in order for such deviation to manifest itself (Assumptions 3.4 and 3.5). Then, the deviation from the globally optimal response happens at some step $K < L$. Furthermore, the conditional probability when generating the token $w_{v_L} \in \mathcal{V}$ is strictly smaller than a positive number, which itself is strictly smaller than 1, i.e.,*

$$\max_{w \in \mathcal{V}} g(X_L = w \mid X_{1:L-1} = w_{\widehat{\mathbf{v}}_T[1]} \ldots w_{\widehat{\mathbf{v}}_T[L-1]}, X_{\leq 0}) < \epsilon_L,$$

$$\text{and } \epsilon_L = \frac{f(X_{1:L} = w_{\mathbf{v}_T^*[1]} \ldots w_{\mathbf{v}_T^*[L-1]} w_{\mathbf{v}_T^*[L]} \mid X_{\leq 0})}{f(X_{1:L-1} = w_{\widehat{\mathbf{v}}_T[1]} \ldots w_{\widehat{\mathbf{v}}_T[L-1]} \mid X_{\leq 0})} < 1.$$

Theorem 3.6 provides a necessary (but not sufficient) condition for the deviation of the stepwise optimal length-$T$ response from the same-length globally optimal response. The uncertainty (i.e., low conditional probabilities) in generating the next token can result from different factors. For

---

[5]Here, we implicitly assume that the context length of LLM is sufficiently large to allow for a meaningful discussion.

[6]When $T = 1$, i.e., if the response is of a length 1, the stepwise optimal is just the globally optimal for an oracle LLM, since there is only one step in total, and $f(X_1 \mid X_{\leq 0}) = g(X_1 \mid X_{\leq 0})$.

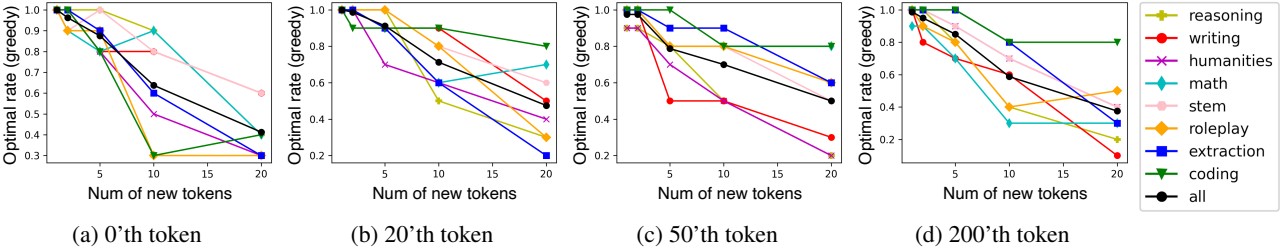

(a) 0'th token      (b) 20'th token      (c) 50'th token      (d) 200'th token

Figure 4: The probability that greedy decoding can attain globally optimal response, with respect to the number of newly generated tokens, and with different starting positions in the generation history. The legend is shared across sub-figures.

instance, a previous mistake or detour makes it challenging to continue in any way that could possibly satisfy the goal specified by the prompt. Such uncertainty can also result from multiple valid ways to proceed in order to achieve the goal. Although we do not have access to the ground-truth conditional probability $f(X_{1:T} \mid X_{\leq 0})$, Theorem 3.6 suggests that when noticeable uncertainty arises, one should to be cautious of a potential deviation from the globally optimal response in the generated text.

**Remark: Implication of Our Theoretical Analysis** Our Theorem 3.6 indicate that, *even if* with an oracle LLM (Assumption 3.3), there is still suboptimality in the purely autogressive way of decoding. In other words, even if the LLM itself perfectly decomposes the (conditional) probabilities (which is a far-fetched benefit to assume in practice), there is still no guarantee in obtaining the globally optimal sequence with purely autoregressive decoding. Our empirical evaluations (Sections 4–5) do not rely on or employ this assumption.

## 4. Sanity Check: Semi-Synthetic Settings

The implication of our theoretical analysis is straightforward. However, it is natural to ask whether the phenomenon actually occurs in real-world LLM decoding scenarios. To provide clear empirical evidence accompanying our theoretical analysis, in this section, we present semi-synthetic experiments that serves as a sanity check. In particular, in moderately realistic settings, we show that greedy decoding for text generation with stepwise optimization results in suboptimal responses. We first outline the semi-synthetic setting, and then present the empirical findings.

**Illustrative Approximation** For any modern LLM with a vocabulary size $|\mathcal{V}|$ (typically on the order of $10^4$ to $10^5$), identifying the globally optimal sequence across multiple steps becomes computationally intractable, even for relatively short sequence lengths ($< 100$). To ensure the validity of our claim while providing a clear and accessible illustration, we adopt beam search as an approximation

strategy of obtaining globally optimal sequence. Since we measure the chance that greedy decoding can attain the global optimum with the stepwise optimal response, this approximation serves as an upper bound on achievable performance, indicating the discrepancy between greedy decoding and the true globally optimal response.

**Approximating Natural Language Scenarios** Since the prompt or context of the generation influence model behavior, we align our experimental setting with common human-LLM interactions. Specifically, we utilize MT-Bench (Zheng et al., 2023) questions as curated prompts, which are designed to evaluate conversational chat models. These samples serve as an approximation of real-world natural language context distributions, ensuring that our findings are grounded in practical scenarios.

**Findings** We use Llama-3.1-8B-Instruct (Llama Team, 2024), and for each prompt, together with a certain length of generation history (0 means only the prompt is given), we evaluate whether the joint probability of the sequence generated with greedy decoding is greater than or equal to that produced by beam search (with beam width set to 10, as a proxy of the global optimum). This comparison indicates the extent to which greedy decoding deviates from the globally optimal response. As illustrated in Figure 4(a), greedy decoding consistently results in suboptimal sequences, and the phenomenon can be observed with a small number of newly generated tokens.

In addition, the potential deviation may behave differently across various positions in the generated text. For instance, when openings of response diverge, it is hard for greedy decoding to achieve optimal results afterwards. To reduce potential inductive bias resulting from the diversity at early stages of generation, we evaluate generations starting/continuing from various positions throughout generation history, as presented in Figures 4(b)–4(d). We can observe that the deviation persists across different positions, which empirically demonstrate the common existence of sub-optimality in autoregressive decoding for text generation.

**Algorithm 1:** Reflection-Window Decoding with Selective Refinement for Text Generation

**Input** : The prompt $X_{\leq 0}$, the reflection window size $d$, as well as hyperparameter-enclosed functions for the pause criterion $\mathrm{IfPause}(\cdot)$ and the refine/correct method $\mathrm{ReGenerate}(\cdot)$.

**Output** : Text generated with selective refinement $\mathbf{x}$.

1   $t^{(\mathrm{slow})} \leftarrow 0, t^{(\mathrm{fast})} \leftarrow 1$; `// slow/fast pointer`

2   response $\mathbf{x} \leftarrow$ empty string;

3   **While** *not stopped, or response not completed* **Do**

4      regular decoding for the next token $w_{t^{(\mathrm{fast})}}$;

5      **If** $t^{(\mathrm{fast})} - t^{(\mathrm{slow})} < d$ **Then**

6        $\mathbf{x} \leftarrow \big(\mathbf{x}_{1:t^{(\mathrm{fast})}-1}, w_{t^{(\mathrm{fast})}}\big)$;

7      **Else**

8        concatenate and get temporary sequence $\widehat{\mathbf{x}} \leftarrow \big(\mathbf{x}_{1:t^{(\mathrm{fast})}-1}, w_{t^{(\mathrm{fast})}}\big)$;

9        **If** $True = \mathrm{IfPause}\big(\widehat{\mathbf{x}}_{t^{(\mathrm{fast})}-d+1:t^{(\mathrm{fast})}}\big)$ **Then**

          ▷ `Refine upon reflection`

10          obtain a length-$d$ replacement text $\mathbf{x}^*_{t^{(\mathrm{fast})}-d+1:t^{(\mathrm{fast})}} \leftarrow \mathrm{ReGenerate}\big(\mathbf{x}_{1:t^{(\mathrm{fast})}-d}\big)$;

11          $\mathbf{x} \leftarrow \big(\mathbf{x}_{1:t^{(\mathrm{fast})}-d}, \mathbf{x}^*_{t^{(\mathrm{fast})}-d+1:t^{(\mathrm{fast})}}\big)$;

12          $t^{(\mathrm{slow})} \leftarrow t^{(\mathrm{fast})}$; `// update slow ptr`

13        **Else**

          ▷ `Continue to generate`

14          $\mathbf{x} \leftarrow \widehat{\mathbf{x}}$;

15      $t^{(\mathrm{fast})} \leftarrow t^{(\mathrm{fast})} + 1$;    `// update fast ptr`

16 **Return**`(x)`.

---

# 5. Empirical Approach and Experiments

In Sections 5.1–5.2, we provide technical details about our empirical approach and settings of our experiments. Then in Sections 5.3–5.4, we present experimental results to demonstrate both the effectiveness and efficiency of our method.

## 5.1. Reflection-Window Decoding: Technical Details

Our findings through both the theoretical characterization of sub-optimality in autoregressive decoding for text generation (Section 3), and the sanity check with empirical verifications in semi-synthetic settings (Section 4), suggest the necessity of a built-in reflection-and-refine mechanism at the decoding level. To empirically address this issue, we propose a selective refinement framework that interchangeably refine and generate as the response unfolds.

Text typically unfolds in a single direction, i.e., from the start to the end, with words, phrases, and sentences. This differentiates text from other forms of objects that occupy multiple dimensional spaces, e.g., images or videos. Taking advantage of this one-dimensional nature, our decoding framework integrates a sliding reflection window along with

two additional modules: (1) a *pausing criterion* that specifies whether we should pause the generation upon reflecting on generated content, and (2) a *refinement/correction method* that facilitates revision at the decoding level (if the pausing criterion is triggered). We present the pseudocode of our reflection-window decoding approach in Algorithm 1.

**Pausing Criterion**   Guided by our theoretical characterization (Theorem 3.6), the reflection at the decoding level needs to capture the (increasing trend of) uncertainty as text generation proceeds. For an empirical pausing criterion, we use the conditional entropy $H(\cdot)$ based on the next-token logits across the vocabulary. Specifically, given an LLM which models the conditional distribution $g(X_t \mid X_{1:t-1})$ of the token at $t$-th step given all the observed history $X_{1:t-1} = \mathbf{x}_{1:t-1}$, we use the pausing criterion $h(t; \sigma, d)$:

$$h(t; \sigma, d) = \begin{cases} \text{True} & \text{if } H\big(X_{t-i} \mid X_{1:(t-i-1)}\big) > \sigma\,, \\ & \forall i \in [0, d-1], \\ \text{False} & \text{Else,} \end{cases} \tag{4}$$

where $\sigma$ denotes the hyperparameter for the threshold of conditional entropy, and $d$ denotes that for the window size (how far we look back in history, in terms of the token counts).

When $h(t^{(\mathrm{fast})}; \sigma, d)$ is True, the pausing criterion (denoted by $\mathrm{IfPause}(\cdot)$ in Algorithm 1, with hyperparameter enclosed) is triggered upon reflecting on the most recent $d$ generated tokens, i.e., the length-$d$ reflection window when the fast pointer is at $t^{(\mathrm{fast})}$. The two parameters, $\sigma$ and $d$, jointly decide the sensitivity and effective region of the pausing criterion, and we present more discussions in Section 5.4.

**Refinement/Correction Method**   When the pausing criterion is triggered, tokens within the current sliding reflection window need to be refined or corrected. Since beam search can approximate the global optimum relatively well, empirically we introduce a fixed-length beam search to generate a new segment with a length $d$ (denoted by the hyperparameter-enclosed function $\mathrm{ReGenerate}(\cdot)$ in Algorithm 1), to replace the content within the reflection window. After the refinement, the slow pointer $t^{(\mathrm{slow})}$ catches up with the fast one $t^{(\mathrm{fast})}$ and the model proceeds with generation while maintaining the sliding reflection window.

**Remark: Versatile Decoding Framework**   We would like to note that our reflection-window decoding approach is highly versatile. While our empirical approach employs a specific pausing criterion and refinement/correction method, practitioners can customize these components by incorporating different functions, namely, $\mathrm{IfPause}(\cdot)$ and $\mathrm{ReGenerate}(\cdot)$ in Algorithm 1, to meet their needs. Our selective refinement framework integrates the sliding reflection window mechanism with these components, enabling simultaneous refinement and generation at the decoding

level while retaining the flexibility to incorporate additional strategies, such as those based on high-level model behaviors and/or speculative decoding (Section 2.2).

## 5.2. Experiment Settings

We provide technical details about settings of our experiments, including models, benchmarks, evaluation metrics, and baseline methods.

**LLM Models** We conduct experiments using multiple models across different families/herds. Specifically, we use Llama-3.1-8B-Instruct (denoted as Llama3.1-8B), which belongs to the Llama 3.1 herds (Llama Team, 2024), Phi-3-Medium-128K-Instruct (Abdin et al., 2024b) (denoted as Phi3-Medium) with 14 billion parameters, Qwen2.5-14B-Instruct (Yang et al., 2024a) (denoted as Qwen2.5-14B) with 14 billion parameters, Qwen2.5-7B-Instruct (denoted as Qwen2.5-7B) with 7 billion parameters, and Mistral-Nemo-Instruct-2407 (MistralAI, 2024) (denoted as Mistral-Nemo) with 12 billion parameters.

**Benchmarks and Evaluation Metrics** Our experiments are conducted on MMLU (Hendrycks et al., 2020) and MT-Bench (Zheng et al., 2023). MMLU tests model's general knowledge across 57 diverse subjects, e.g., humanities, STEM (Science, Technology, Engineering, and Mathematics), and social sciences, at varying difficulty levels, making it a comprehensive evaluation of model's reasoning performance and factual knowledge. MT-Bench, on the other hand, provides a fine-grained evaluation through multi-turn conversational tasks, evaluating not just correctness, but also coherence and fluency.

For MMLU, we adopt the macro averaged accuracy metric because the number of questions varies across subjects. For MT-Bench, for each pair of responses, we prompt the LLM judge (for which we use GPT-4o, OpenAI 2024) with two responses following the prompting method outlined in Zheng et al. (2023). The LLM judge should return a decision from three options: win, lose or tie. To avoid the influence from the preference bias, for each pair we prompt the LLM judge twice with responses placed in different sequences. When a response gets two win's (lose's) or one win (lose) plus one tie, we record the response as (not) prevailing. The rest situations are treated as tie. We use win rate as the measurement of performance on MT-Bench, calculated by win rate $:= \frac{\text{number of wins}}{\text{number of wins}+\text{number of losses}}$.

**Baseline Methods** We compare reflection-window decoding with three baseline methods: greedy decoding, vanilla beam search (with a finite beam width), and Top-$k$/Top-$p$

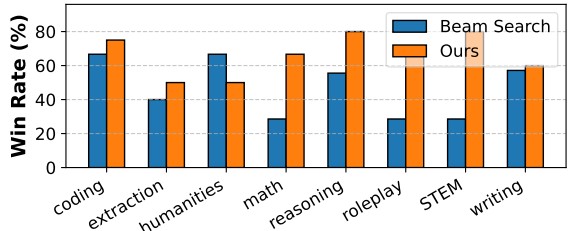

Figure 5: Comparison of win rates between beam search and our reflection-window decoding (both against greedy decoding) on MT-Bench across categories

sampling (Fan et al., 2018; Holtzman et al., 2020).[7] We consider greedy search as one baseline method since it corresponds closely to our theoretical analysis. However, since the widely usage of Top-$p$ and Top-$k$ sampling (and often used in combination), we also include it as a baseline approach. Reflection-window decoding only leverages beam search when necessary, i.e., when IfPause$(\cdot)$ in Algorithm 1 is triggered. For fair comparison, we set the beam size to 4 in both reflection-window decoding and the vanilla beam search.

## 5.3. Experiment Results

We compare the performance among greedy decoding, beam search, Top-$k$/Top-$p$ sampling, and our reflection-window decoding. We use an entropy threshold of $\sigma = 0.5$ and a window size of $d = 4$ in reflection-window decoding.

**MMLU** Table 1 presents the comparison among greedy decoding, beam search, and reflection-window decoding (with greedy search as the "regular decoding" in Algorithm 1) on four question categories in MMLU (STEM, humanities, social sciences, and others), using Phi3-Medium as the base LLM. Our approach demonstrates comparable or superior performance across all categories, achieving the highest average accuracy of 78.15%. The results containing all MMLU subjects can be found in Table 11 in Appendix.

**MT-Bench** We choose Llama3.1-8B as the base model in this experiment. On MT-Bench, the Reflection Window method clearly outperformed both greedy decoding and beam search. In comparisons with greedy decoding, according to assessments by GPT-4o, reflection window prevails in 66.67% of cases, a win rate significantly higher than that of beam search, which only outperformed greedy decoding in 46.3% of cases.

---

[7] If there is no limit on the computation and storage, beam search with an unconstrained beam width could yield the globally optimal output through brute force. However, in practice, maintaining a full frontier quickly becomes intractable, and a finite beam width is often introduced as a hyperparameter.

Table 1: Comparison of macro averaged accuracy on MMLU across subject categories with Phi3-Medium

| Decoding Method | STEM | Humanities | Social Sciences | Others | Macro Average |
|---|---|---|---|---|---|
| Greedy Decoding | 78.40% | 71.92% | **83.91%** | **81.59%** | 78.14% |
| Beam Search | 78.62% | 68.65% | 82.93% | 79.59% | 76.44% |
| Reflection-Window (Greedy) | **79.06%** | **71.98%** | 83.65% | 81.10% | **78.15%** |

Table 2: Summary of comparison between Top-$k$/Top-$p$ and reflection-window decoding on MMLU (multiple models)

| Model | Decoding Method | STEM | Humanities | Social Sciences | Others | Average |
|---|---|---|---|---|---|---|
| Llama3.1-8B | Top-$k$ Top-$p$ | 65.94% | 58.43% | 72.60% | 73.06% | 66.46% |
| | Reflection-Window | **67.21%** | **59.43%** | **72.86%** | **73.41%** | **67.21%** |
| Phi3-Medium | Top-$k$ Top-$p$ | 71.36% | 65.76% | 79.10% | **74.93%** | 71.97% |
| | Reflection-Window | **73.07%** | **71.31%** | **79.85%** | 74.86% | **74.73%** |
| Qwen2.5-14B | Top-$k$ Top-$p$ | 83.00% | 69.86% | 82.74% | 79.82% | 77.84% |
| | Reflection-Window | **83.30%** | **70.33%** | **83.85%** | **81.41%** | **78.48%** |
| Mistral-Nemo | Top-$k$ Top-$p$ | 60.74% | 50.16% | 67.79% | **65.66%** | 59.83% |
| | Reflection-Window | **61.78%** | **51.41%** | **69.22%** | 65.27% | **60.71%** |

Table 3: Regeneration metrics on MMLU with Llama3.1-8B

| Category | ReGen. Ratio (%) | ReGen. Call |
|---|---|---|
| STEM | 3.50 | 3.15 |
| Humanities | 5.04 | 4.27 |
| Social Sciences | 4.82 | 3.84 |
| Others | 5.54 | 4.31 |

Table 4: Acc. by $\sigma$ (MMLU Social Sciences, Qwen2.5-7B)

| $\sigma$ | 0.1 | 0.25 | 0.5 | 0.75 | 1.0 |
|---|---|---|---|---|---|
| Acc. (%) | 79.40 | **80.31** | 79.82 | 79.88 | 79.69 |

Table 5: Acc. by $d$ (MMLU Social Sciences, Qwen2.5-7B)

| $d$ | 2 | 3 | 4 | 5 | 6 |
|---|---|---|---|---|---|
| Acc. (%) | 79.66 | 79.75 | 79.82 | 79.62 | **79.88** |

Figure 5 shows that our reflection-window decoding significantly outperforms beam search in Roleplay, STEM, Math, and Reasoning tasks. These tasks require strong logical consistency, and greedy or fixed search strategies often lead to early sub-optimal choices that degrade output quality. Our approach mitigates this by enabling refinement during generation, making generated contents more coherent.

**Compatibility with Top-$k$/Top-$p$ Sampling**  Reflection-window decoding generates tokens autoregressively when the pausing criterion is not triggered and only performs selective refinements. The framework is compatible with Top-$k$/Top-$p$ sampling except in the refinement/correction mode. In these experiments, we set $k = 10$, $p = 0.9$, and temperature as 1.0 for both our approach and the baseline Top-$k$/Top-$p$ sampling. As shown in Table 2, our decoding approach consistently outperforms the standard Top-$k$/Top-$p$ approach across all four models. In particular, the average accuracy

improvements range from $0.88$ percentage points (Mistral-Nemo) to $2.76$ percentage points (Phi3-Medium), highlighting the effectiveness of our approach even when stochastic sampling is introduced. Notably, the largest performance gains are observed in STEM and humanities categories, suggesting that reflection-window decoding is particularly beneficial for reasoning-heavy tasks. This aligns with observations from the MT-Bench experiments, which also demonstrate that our approach excels in tasks demanding structured logical thinking and complex problem-solving. We present further results from Table 12 to Table 15 in Appendix.

### 5.4. Further Discussions and Analyses

**Efficiency of Reflection-Window Decoding**  In Table 3, we aggregate the regeneration statistics on MMLU with Llama3.1-8B. We record two metrics: (1) the regeneration ratio, which calculates the overall ratio of refined/corrected tokens in the completed response, and (2) the regeneration call, which counts the number of times the pausing criterion is triggered and refinement/correction is needed before finishing any particular response. We find that regeneration ratio ranges from $3.5\%$ to $5.5\%$ across all categories with a relatively mild pausing criterion. This suggests that the refinements are usually needed during decoding. While beam search always maintains a complete frontier of candi-

date sequences, our reflection-window decoding approach only activates beam search when necessary, and at the sub-sequence level.

**Entropy Threshold $\sigma$ and Window Size $d$**  We investigate the impact of threshold $\sigma$ on MMLU Social Sciences with Qwen2.5-7B with fixed window size $d = 4$. The results in Table 4 demonstrate that our method performs robustly across $\sigma$ values ranging from $0.25$ to $0.75$, with $\sigma = 0.25$ achieving the highest macro average of $80.31\%$. While our default setting of $\sigma = 0.5$ is not the best in this specific experiment, it maintains strong performance and shows consistent improvements on most subjects, suggesting that it serves as a reliable default configuration for general usage.

In terms of window size $d$, we also choose social science subjects using Qwen2.5-7B. The results in Table 5 show that our method maintains strong performance across various window sizes ($d = 2$ to $d = 6$), with overall macro averages consistently around $79.7\%$. While $d = 6$ achieves the highest macro average, $d = 4$ demonstrates comparable performance and maintains better computational efficiency. These results further support our choice of $d = 4$ as a robust default setting, offering a good balance between performance and efficiency across different models and tasks. We provide additional results on MT-Bench in Table 6 and Table 7, and on MMLU in Table 8 and Table 9 in Appendix.

## 6. Conclusion

In this paper, we theoretically characterize one inherent shortcoming, among others, of the autoregressive decoding for text generation in LLMs. In particular, we show that even when the optimality is defined in terms of the joint probability over all generated tokens, an oracle LLM can still potentially deviate from the globally optimal response of the same length. To mitigate the sub-optimality of the autoregressive way of text generation, we propose an empirical approach guided by our theoretical characterization. We incorporate a sliding reflection window and a pausing criterion so that refinement and generation can be performed interchangeably. Our experimental results demonstrate that our reflection-window decoding strategy achieves significant improvement over regular decoding strategies in inference-intensive settings and maintains performance that is comparable, or even superior to, beam search while being more efficient.

## Acknowledgments

We would also like to acknowledge the support from NSF Award No. 2229881, AI Institute for Societal Decision Making (AI-SDM), the National Institutes of Health (NIH) under Contract R01HL159805, and grants from Quris AI, Florin Court Capital, and MBZUAI-WIS Joint Program. ZT is supported by the National Institute of Justice (NIJ) Graduate Research Fellowship, Award No. 15PNIJ-24-GG-01565-RESS.

## Impact Statement

In this paper, we theoretically characterize and empirically address the sub-optimality of the autoregressive decoding for text generation. In particular, we propose a selective refinement framework and implement it with a sliding reflection window mechanism, enabling interchangeable refinement and generation as the decoding proceeds. Our approach strikes a balance between efficiency and optimality. There are many potential societal consequences of our work, none which we feel must be specifically highlighted here.

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

# Supplement to "Reflection-Window Decoding: Text Generation with Selective Refinement"

## Table of Contents: Appendix

## List of Tables

## A. The Proof of Theoretical Result

**Theorem A.1** (Indication of Deviation from the Globally Optimal Length-$T$ Response). *Given the prompt $X_{\leq 0}$, when an oracle LLM (Assumption 3.3) generates a stepwise optimal length-$T$ response which is not the globally optimal response with the same length, let $L \leq T$ denote the minimum length of prefix-sequence needed in order for such deviation to manifest itself (Assumptions 3.4 and 3.5). Then, the deviation from the globally optimal response happens at some step $K < L$. Furthermore, the conditional probability when generating the token $w_{v_L} \in \mathcal{V}$ is strictly smaller than a positive number, which itself is strictly smaller than 1, i.e.,*

$$1 > \epsilon_L > \max_{w \in \mathcal{V}} g(X_L = w \mid X_{1:L-1} = w_{\widehat{\mathbf{v}}_T[1]} w_{\widehat{\mathbf{v}}_T[2]} \ldots w_{\widehat{\mathbf{v}}_T[L-1]}, X_{\leq 0}),$$

$$\text{where } \epsilon_L = \frac{f(X_{1:L} = w_{\mathbf{v}_T^*[1]} w_{\mathbf{v}_T^*[2]} \ldots w_{\mathbf{v}_T^*[L-1]} w_{\mathbf{v}_T^*[L]} \mid X_{\leq 0})}{f(X_{1:L-1} = w_{\widehat{\mathbf{v}}_T[1]} w_{\widehat{\mathbf{v}}_T[2]} \ldots w_{\widehat{\mathbf{v}}_T[L-1]} \mid X_{\leq 0})}. \tag{5}$$

*Proof.* We first show that the deviation from the globally optimal response happens before step $L$. Then, we show that the conditional probability when generating the token $w_{v_L}$ is bounded away from 1.

By definition of oracle LLM (Assumption 3.3), the advantage of the globally optimal response cannot manifest itself at $L = 1$ (even if the deviation happens at step 1), i.e., $L > 1$. Since the minimum length of prefix-sequence needed in order for the deviation of stepwise optimal response from the same-length globally optimal response to manifest is $L$, then the advantage of the globally optimal response is not manifested until step $L$. Until step $L - 1$, in terms of the ground-truth conditional probability following the prompt $X_{\leq 0}$, prefix-sequences of the globally optimal response is *not* strictly preferred compared to their same-length counterparts of the stepwise optimal response:

$$f(X_1 = w_{\widehat{\mathbf{v}}_T[1]} \mid X_{\leq 0}) \geq f(X_1 = w_{\mathbf{v}_T^*[1]} \mid X_{\leq 0}),$$

$$f(X_{1:2} = w_{\widehat{\mathbf{v}}_T[1]} w_{\widehat{\mathbf{v}}_T[2]} \mid X_{\leq 0}) \geq f(X_{1:2} = w_{\mathbf{v}_T^*[1]} w_{\mathbf{v}_T^*[2]} \mid X_{\leq 0}),$$

$$\ldots$$

$$f(X_{1:L-1} = w_{\widehat{\mathbf{v}}_T[1]} w_{\widehat{\mathbf{v}}_T[2]} \ldots w_{\widehat{\mathbf{v}}_T[L-1]} \mid X_{\leq 0}) \geq f(X_{1:L-1} = w_{\mathbf{v}_T^*[1]} w_{\mathbf{v}_T^*[2]} \ldots w_{\mathbf{v}_T^*[L-1]} \mid X_{\leq 0}). \tag{6}$$

Starting from step $L$ and onwards (Assumption 3.5), prefix-sequences of the globally optimal response are strictly preferred compared to their counterparts of the stepwise optimal response:

$$f(X_{1:L} = w_{\widehat{\mathbf{v}}_T[1]} \ldots w_{\widehat{\mathbf{v}}_T[L-1]} w_{\widehat{\mathbf{v}}_T[L]} \mid X_{\leq 0}) < f(X_{1:L} = w_{\mathbf{v}_T^*[1]} \ldots w_{\mathbf{v}_T^*[L-1]} w_{\mathbf{v}_T^*[L]} \mid X_{\leq 0}),$$

$$\ldots$$

$$f(X_{1:T} = w_{\widehat{\mathbf{v}}_T[1]} w_{\widehat{\mathbf{v}}_T[2]} \ldots w_{\widehat{\mathbf{v}}_T[T]} \mid X_{\leq 0}) < f(X_{1:T} = w_{\mathbf{v}_T^*[1]} w_{\mathbf{v}_T^*[2]} \ldots w_{\mathbf{v}_T^*[T]} \mid X_{\leq 0}). \tag{7}$$

Assumption 3.4 specifies that for any two same-length but different sequences following the prompt $X_{\leq 0}$, there is a strict ordering between them. Then, in order for the advantage of the globally optimal length-$T$ response to manifest, in terms of strict preferences staring from the length-$L$ prefix-sequence (Equation (7)), there is *at least one* strict preference of the prefix-sequence of stepwise optimal response over its globally optimal counterpart before step $L$. In other words, there is at least one step $K \in [1, L-1]$ such that a strict preference ("$>$" instead of "$\geq$") is present in Equation (6):

$$f(X_{1:K} = w_{\widehat{\mathbf{v}}_T[1]} w_{\widehat{\mathbf{v}}_T[2]} \ldots w_{\widehat{\mathbf{v}}_T[K]} \mid X_{\leq 0}) > f(X_{1:K} = w_{\mathbf{v}_T^*[1]} w_{\mathbf{v}_T^*[2]} \ldots w_{\mathbf{v}_T^*[K]} \mid X_{\leq 0}). \tag{8}$$

In order to see why this is the case, consider the opposite scenario where there is *no* strict preference in Equation (6). Under Assumption 3.4, the comparison between prefix-sequences is either strict preference (they are different) or exactly the same (identical sequences). If there is no strict preference in Equation (6), then for all $t \in [1, L-1]$, $w_{\widehat{\mathbf{v}}_T[t]} = w_{\mathbf{v}_T^*[t]}$, i.e., the first $L - 1$ tokens in the stepwise optimal response are the length-$(L-1)$ prefix of the globally optimal response. If this is the case, the token generated at step $L$ has to deviate from the globally optimal response (since $L$ is the minimum length for

the deviation to manifest) $w_{\widehat{\mathbf{v}}_T[L]} \neq w_{\mathbf{v}_T^*[L]}$:

$$
\begin{aligned}
&f(X_{1:L} = w_{\widehat{\mathbf{v}}_T[1]} \ldots w_{\widehat{\mathbf{v}}_T[L-1]} w_{\widehat{\mathbf{v}}_T[L]} \mid X_{\leq 0}) \\
&\overset{(i)}{=} g(X_L = w_{\widehat{\mathbf{v}}_T[L]} \mid X_{1:L-1} = w_{\widehat{\mathbf{v}}_T[1]} \ldots w_{\widehat{\mathbf{v}}_T[L-1]}, X_{\leq 0}) \cdot f(X_{1:L-1} = w_{\widehat{\mathbf{v}}_T[1]} \ldots w_{\widehat{\mathbf{v}}_T[L-1]} \mid X_{\leq 0}) \\
&\overset{(ii)}{=} g(X_L = w_{\widehat{\mathbf{v}}_T[L]} \mid X_{1:L-1} = w_{\mathbf{v}_T^*[1]} \ldots w_{\mathbf{v}_T^*[L-1]}, X_{\leq 0}) \cdot f(X_{1:L-1} = w_{\mathbf{v}_T^*[1]} \ldots w_{\mathbf{v}_T^*[L-1]} \mid X_{\leq 0}) \\
&\overset{(iii)}{>} g(X_L = w_{\mathbf{v}_T^*[L]} \mid X_{1:L-1} = w_{\mathbf{v}_T^*[1]} \ldots w_{\mathbf{v}_T^*[L-1]}, X_{\leq 0}) \cdot f(X_{1:L-1} = w_{\mathbf{v}_T^*[1]} \ldots w_{\mathbf{v}_T^*[L-1]} \mid X_{\leq 0}) \\
&\overset{(iv)}{=} f(X_{1:L} = w_{\mathbf{v}_T^*[1]} \ldots w_{\mathbf{v}_T^*[L-1]} w_{\mathbf{v}_T^*[L]} \mid X_{\leq 0}),
\end{aligned}
\tag{9}
$$

where $(i)$ and $(iv)$ follow Assumption 3.3, $(ii)$ corresponds to the setting in this opposite scenario, and $(iii)$ follows Definition 3.2 and that $w_{\widehat{\mathbf{v}}_T[L]} \neq w_{\mathbf{v}_T^*[L]}$. This preference relation in Equation (9) contradicts with that in Equation (7), and therefore, Equation (8) has to hold true.

Therefore, when the advantage of the globally optimal response does not manifest itself until step $L$, the stepwise optimal response deviates from the globally optimal counterpart at some step $K < L$, and that under Assumption 3.4, the following strict preference relations hold true:

$$
\begin{aligned}
f(X_{1:K} = w_{\widehat{\mathbf{v}}_T[1]} w_{\widehat{\mathbf{v}}_T[2]} \ldots w_{\widehat{\mathbf{v}}_T[K]} \mid X_{\leq 0}) &> f(X_{1:K} = w_{\mathbf{v}_T^*[1]} w_{\mathbf{v}_T^*[2]} \ldots w_{\mathbf{v}_T^*[K]} \mid X_{\leq 0}), \\
&\cdots \\
f(X_{1:L-1} = w_{\widehat{\mathbf{v}}_T[1]} w_{\widehat{\mathbf{v}}_T[2]} \ldots w_{\widehat{\mathbf{v}}_T[L-1]} \mid X_{\leq 0}) &> f(X_{1:L-1} = w_{\mathbf{v}_T^*[1]} w_{\mathbf{v}_T^*[2]} \ldots w_{\mathbf{v}_T^*[L-1]} \mid X_{\leq 0}).
\end{aligned}
\tag{10}
$$

This, together with Equation (7) and Assumption 3.3, indicates that:

$$
\begin{aligned}
&g(X_L = w_{\widehat{\mathbf{v}}_T[L]} \mid X_{1:L-1} = w_{\widehat{\mathbf{v}}_T[1]} \ldots w_{\widehat{\mathbf{v}}_T[L-1]}, X_{\leq 0}) \\
&\overset{(i)}{=} \frac{f(X_{1:L} = w_{\widehat{\mathbf{v}}_T[1]} \ldots w_{\widehat{\mathbf{v}}_T[L-1]} w_{\widehat{\mathbf{v}}_T[L]} \mid X_{\leq 0})}{f(X_{1:L-1} = w_{\widehat{\mathbf{v}}_T[1]} \ldots w_{\widehat{\mathbf{v}}_T[L-1]} \mid X_{\leq 0})} \\
&\overset{(ii)}{<} \frac{f(X_{1:L} = w_{\mathbf{v}_T^*[1]} \ldots w_{\mathbf{v}_T^*[L-1]} w_{\mathbf{v}_T^*[L]} \mid X_{\leq 0})}{f(X_{1:L-1} = w_{\widehat{\mathbf{v}}_T[1]} \ldots w_{\widehat{\mathbf{v}}_T[L-1]} \mid X_{\leq 0})} = \epsilon_L \\
&\overset{(iii)}{<} \frac{f(X_{1:L} = w_{\mathbf{v}_T^*[1]} \ldots w_{\mathbf{v}_T^*[L-1]} w_{\mathbf{v}_T^*[L]} \mid X_{\leq 0})}{f(X_{1:L-1} = w_{\mathbf{v}_T^*[1]} w_{\mathbf{v}_T^*[2]} \ldots w_{\mathbf{v}_T^*[L-1]} \mid X_{\leq 0})} \\
&\overset{(iv)}{=} g(X_L = w_{\mathbf{v}_T^*[L]} \mid X_{1:L-1} = w_{\mathbf{v}_T^*[1]} \ldots w_{\mathbf{v}_T^*[L-1]}, X_{\leq 0}) \leq 1,
\end{aligned}
\tag{11}
$$

where $(i)$ and $(iv)$ follow Assumption 3.3, $(ii)$ follows Equation (7), and $(iii)$ follows Equation (10).

Therefore, the conditional probability of generating any $w_{v_L}$ is strictly smaller than a positive number $\epsilon_L$, which is further strictly smaller than a positive number upper-bounded by 1. $\qquad\square$

# B. Additional Results and Analyses

In this section, we present additional results and further discussions on influences from hyperparameters. We also provide concrete examples demonstrating the process and overall performance of our reflection-window decoding.

## B.1. Analysis on Window Size $d$

We conduct extensive experiments on MT-Bench to analyze the impact of window size using both Mistral-Nemo (Table 6) and Llama3.1-8B (Table 7), with a fixed pausing criterion with $\sigma = 0.5$. These GPT-4o evaluator scores on MT-Bench provide additional evidence that our approach consistently outperforms traditional decoding methods.

For Mistral-Nemo, the optimal performance is achieved at $d = 3$ with an overall score of 7.93, surpassing both greedy decoding and beam search. For Llama3.1-8B, our method consistently outperforms both greedy decoding and beam search across different window sizes, with $d = 5$ achieving the best overall performance. While $d = 4$ may not always yield the best result, it demonstrates robust performance across both models and serves as a reliable default setting.

We further evaluate how different window sizes affect the performance on MMLU social science tasks using Qwen2.5-7B. The results are presented in Table 8.

Table 6: Performance across different window sizes $d$ on MT-Bench with Mistral-Nemo

| Decoding Method | Rating 1 | Rating 2 | Overall Mean |
|---|---|---|---|
| Reflection-Window ($d = 2$) | 8.38 | 7.28 | 7.82 |
| Reflection-Window ($d = 3$) | **8.44** | 7.42 | **7.93** |
| Reflection-Window ($d = 4$) | 8.28 | 7.41 | 7.84 |
| Greedy Decoding | 8.38 | 7.29 | 7.83 |
| Beam Search | 8.32 | **7.49** | 7.91 |

Table 7: Performance across different window sizes $d$ on MT-Bench with Llama3.1-8B

| Decoding Method | Rating 1 | Rating 2 | Overall Mean |
|---|---|---|---|
| Reflection-Window ($d = 2$) | 8.29 | 7.09 | 7.69 |
| Reflection-Window ($d = 3$) | 8.35 | 7.51 | 7.93 |
| Reflection-Window ($d = 4$) | **8.36** | 7.42 | 7.89 |
| Reflection-Window ($d = 5$) | 8.31 | **7.62** | **7.97** |
| Greedy Decoding | 8.28 | 7.49 | 7.88 |
| Beam Search | 8.07 | 7.19 | 7.63 |

Table 8: Accuracy (%) across different window sizes $d$ on MMLU Social Sciences with Qwen2.5-7B

| Subject | $d = 2$ | $d = 3$ | $d = 4$ | $d = 5$ | $d = 6$ |
|---|---|---|---|---|---|
| Econometrics | 62.28 | 62.28 | **64.91** | 64.04 | **64.91** |
| High School Geography | 85.86 | 84.34 | 86.36 | **87.37** | 85.86 |
| High School Government and Politics | **93.26** | **93.26** | 92.23 | 91.19 | 92.23 |
| High School Macroeconomics | 75.90 | **76.15** | 75.13 | **76.15** | 75.64 |
| High School Microeconomics | 83.61 | **84.03** | 83.61 | 82.77 | 83.19 |
| High School Psychology | 87.89 | 88.07 | 88.07 | 88.07 | **88.26** |
| Human Sexuality | 77.86 | 75.57 | 78.63 | **79.39** | 77.86 |
| Professional Psychology | **73.86** | 73.37 | 73.20 | 72.88 | 73.37 |
| Public Relations | 68.18 | **70.00** | **70.00** | 65.45 | 68.18 |
| Sociology | 71.02 | 73.06 | **73.47** | 72.24 | 73.06 |
| Security Studies | 83.08 | 83.58 | 83.08 | 84.08 | **84.58** |
| US Foreign Policy | 86.00 | 86.00 | 86.00 | 86.00 | **88.00** |
| Macro Average | 79.66 | 79.75 | 79.82 | 79.62 | **79.88** |

**B.2. Analysis on Threshold $\sigma$**

We investigate the impact of threshold $\sigma$ on MMLU social science subjects using Qwen2.5-7B with a fixed window size $d = 4$. The detailed results are presented in Table 9.

Table 9: Accuracy (%) across different entropy thresholds $\sigma$ on MMLU Social Sciences with Qwen2.5-7B

| Subject | $\sigma = 0.1$ | $\sigma = 0.25$ | $\sigma = 0.5$ | $\sigma = 0.75$ | $\sigma = 1.0$ |
|---|---|---|---|---|---|
| Econometrics | 62.28 | **64.91** | **64.91** | **64.91** | **64.91** |
| High School Geography | **92.23** | 91.71 | **92.23** | **92.23** | 91.19 |
| High School Government and Politics | **92.23** | 91.71 | **92.23** | **92.23** | 91.19 |
| High School Macroeconomics | 75.13 | **76.67** | 75.13 | 75.90 | 75.90 |
| High School Microeconomics | **84.45** | **84.45** | 83.61 | 83.19 | 83.61 |
| High School Psychology | 87.52 | **88.44** | 88.07 | 88.26 | 88.07 |
| Human Sexuality | 74.05 | 77.86 | **78.63** | 77.10 | 77.10 |
| Professional Psychology | 73.20 | **74.35** | 73.20 | 73.20 | 73.04 |
| Public Relations | 69.09 | **70.00** | **70.00** | 67.27 | 67.27 |
| Sociology | 84.58 | **85.07** | 83.08 | 84.58 | 84.58 |
| Security Studies | 72.24 | 71.43 | **73.47** | 72.24 | 72.24 |
| US Foreign Policy | **86.00** | 85.00 | 85.00 | 85.00 | **87.00** |
| Macro Average | 79.40 | **80.31** | 79.82 | 79.88 | 79.69 |

**B.3. Analysis on Regeneration Ratio**

To further understand the computational efficiency of our method, we analyze the regeneration ratio under different window size settings. We select six college-level subject categories from the MMLU test set (including biology, chemistry, computer science, mathematics, medicine, and physics) for analysis, and conduct experiments with the Llama3.1-8B model with a threshold of $\sigma = 0.5$. We consider the window size $d$ as the key hyperparameter, because it directly influences the regeneration ratio, which is calculate by the product of the times criterion get triggered (the regeneration call) and the window size ($d$), divided by the total length of final response.

As shown in Table 10 and Figure 6, as the window size increases from 2 to 4, the average regeneration ratio shows a clear downward trend, decreasing from 9.60% to 3.70%. The trend indicates that larger window sizes lead to a faster decrease in the number of modifications needed. Notably, across all settings, the regeneration ratio remains below 15%, suggesting that our method maintains comparable computational workload as greedy decoding for the majority of the time. These results demonstrate the efficiency of our approach, since it only invoke beam search to find optimal approximations for sub-sequences when necessary, while maintaining the overall efficiency comparable to greedy decoding.

Table 10: Average regeneration ratio by window sizes $d$ on MMLU college-level subjects with Llama3.1-8B

| Window Size $d$ | 2 | 3 | 4 |
|---|---|---|---|
| Average Regeneration Ratio (%) | 9.60 | 6.02 | **3.70** |

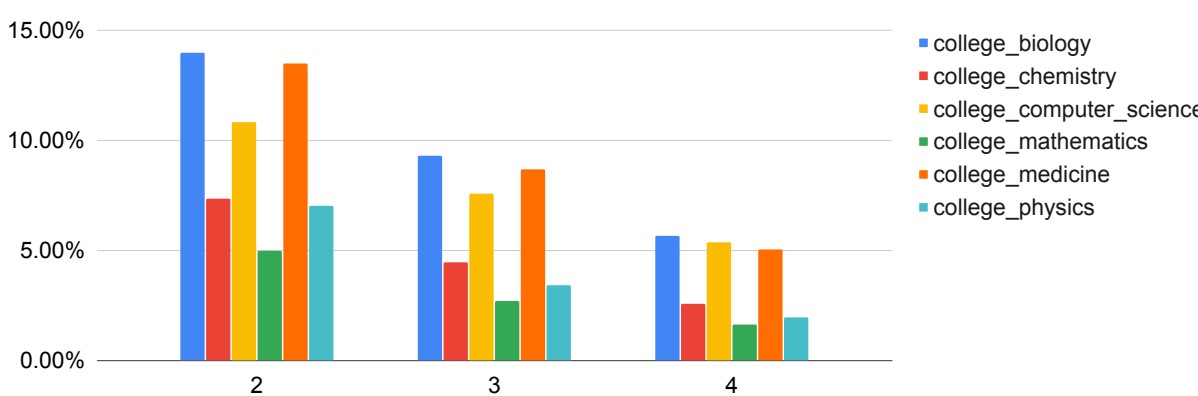

Figure 6: Regeneration ratio on MMLU college-level subjects with Llama3.1-8B

## B.4. Demonstrative Examples on MT-Bench

In this subsection, we present three examples selected from MT-Bench to illustrate the generation process in our reflection-window decoding. The first example is the 28th test sample from MT-Bench. As is shown in Figure 7, the pausing criterion gets triggered when *"part of the car"* was generated. The part gets refined and replaced by *"specific part, whereas"* which leads to a more flexible and natural answer. Figure 8 and Figure 9 are selected from the 29th and 7th test sample of MT-Bench, respectively.

**Question**

**Which word does not belong with the others?** tyre, steering wheel, car, engine

**Reflection Window**

The word that does not belong with the others is "tyre".
The other three words–"steering wheel", "car", and "engine"–are all related to a vehicle, specifically a car.
A "tyre", on the other hand, is a component of a vehicle, but it's a ~~part of the car~~ **specific part, whereas** the other three words are more general terms related to the vehicle as a whole.
However, if I had to choose one word that stands out as not belonging, I would say "steering wheel". The other three words–"car", "engine", and "tyre"–are all nouns that refer to parts or aspects of a vehicle, whereas "steering wheel" is a specific component of a vehicle.
But if I had to choose one word that is the least related to the others, I would say "car". The other three words–"steering wheel", "engine", and "tyre"–are all components or parts of a vehicle, whereas "car" is the vehicle itself.
So, it really depends on how you look at it!

**Greedy Decoding**

The word that does not belong with the others is "tyre".
The other three words–"steering wheel", "car", and "engine"–are all related to a vehicle, specifically a car.
A "tyre", on the other hand, is a component of a vehicle, but it's a part of the car, rather than the car itself.

**Beam Search**

The word that does not belong with the others is "tyre".
The other three words–"steering wheel", "car", and "engine"–are all related to a vehicle, specifically a car.
A "tyre", on the other hand, is a component of a vehicle, but it's not a vehicle itself.

Figure 7: Test Sample #28

## Question

**Could you write a captivating short story beginning with the sentence:**
*The old abandoned house at the end of the street held a secret that no one had ever discovered.*

## Reflection Window

Here's a captivating short story for you:
The old abandoned house at the end of the street held a secret that no one had ever discovered...
...(omitted)
Suddenly, Emily heard a faint whispering in her ear. It was a soft, raspy voice that seemed to be ~~speaking an ancient~~ **coming from all around** her. "Welcome, Emily," it said. "I've been waiting for you."
...(omitted)
From that day on, Emily returned to the old house again and again, learning the secrets of the nexus point and the magic that lay within. And as she did, she discovered that the house was not just a place – it was a doorway to a whole new world of wonder and possibility.

## Greedy Decoding

Here's a captivating short story for you:
The old abandoned house at the end of the street held a secret that no one had ever discovered...
...
Suddenly, Emily heard a faint whispering in her ear. It was a soft, raspy voice that seemed to be speaking an ancient language she couldn't quite understand. But as she listened more closely, she began to pick out words – words that seemed to be calling to her, drawing her deeper into the heart of the house.
...(omitted)
But as she looked up, she saw that she was no longer alone. A figure stood in the doorway, watching her with eyes that seemed to hold a deep sadness and a hint of longing. And as Emily looked into those eyes, she knew that she had stumbled upon a secret that was not just about the house, but about herself – a secret that would change her life forever.

## Beam Search

Here's a captivating short story for you:
The old abandoned house at the end of the street held a secret that no one had ever discovered. For as long as anyone could remember, the house had stood vacant, its grandeur and beauty slowly being consumed by the passing of time.
...(omitted)
It was then that Emily realized that she was not just a curious adventurer, but a key player in a much larger game. The old abandoned house at the end of the street was not just a secret, but a doorway to a world of wonder and discovery, and Emily was about to embark on a journey that would change her life forever.

Figure 8: Test Sample #29

## Question

One morning after sunrise, Suresh was standing facing a pole. The shadow of the pole fell exactly to his right. Can you tell me the direction towards which the shadow was pointing–east, south, west, or north? . . .
**To which direction was Suresh facing? How do you solve this?**

## Reflection Window

To determine the direction Suresh was facing, we need to consider the position of the sun and the direction of the shadow.
Since the ~~shadow of the pole~~ **sun is in the east** and its rays cast shadows to the west, and the shadow of the pole fell to Suresh's right, we can infer that Suresh was facing **south**. This is because, when the sun is in the east, the shadows fall to the west, and if the shadow falls to the right, it means Suresh was facing south.

## Greedy Decoding

To determine the direction in which Suresh was facing, we need to consider the position of the sun and the direction of the shadow.
Since the shadow of the pole fell to Suresh's right, we can infer that the sun is in the east. This is because the sun's rays are coming from the east, and the shadows are cast in the opposite direction. Now, since the shadow fell to Suresh's right, we can conclude that Suresh was facing the **west**. This is because the shadow fell to his right, which means he was facing the opposite direction, i.e., the west.
Therefore, the direction in which Suresh was facing is **west**.

## Beam Search

Since Suresh was standing facing a pole and the shadow of the pole fell exactly to his right, we can deduce the direction of the shadow based on the position of the sun at sunrise.
1. At sunrise, the sun rises in the east.
2. Since Suresh was facing the pole and the shadow of the pole fell to his right, it means the shadow was pointing in a direction perpendicular to his line of sight.
3. Given that the sun rises in the east, the shadow of the pole would fall to the right of Suresh's line of sight, which means the shadow would be pointing **west**.
Therefore, the direction towards which the shadow of the pole was pointing is **west**.

Figure 9: Test Sample #7

## C. Performance Across All MMLU Subjects

Due to the size of the table, the material is arranged in the one-table-per-page manner (starting from the next page).

Table 11: Comparison among greedy decoding, beam search, and reflection-window decoding on MMLU with Phi3-Medium

| Subject | Greedy Decoding | Beam Search | Reflection-Window |
|---|---|---|---|
| abstract_algebra | 58.00 | 56.00 | 59.00 |
| anatomy | 73.33 | 74.81 | 72.59 |
| astronomy | 87.50 | 84.21 | 88.82 |
| business_ethics | 80.00 | 74.00 | 79.00 |
| clinical_knowledge | 84.91 | 83.40 | 84.53 |
| college_biology | 86.81 | 88.19 | 88.19 |
| college_chemistry | 55.00 | 59.00 | 58.00 |
| college_computer_science | 68.00 | 68.00 | 70.00 |
| college_mathematics | 62.00 | 58.00 | 57.00 |
| college_medicine | 78.61 | 75.14 | 75.72 |
| college_physics | 77.45 | 73.53 | 80.39 |
| computer_security | 80.00 | 75.00 | 79.00 |
| conceptual_physics | 80.85 | 82.98 | 82.55 |
| econometrics | 59.65 | 57.89 | 61.40 |
| electrical_engineering | 70.34 | 68.97 | 72.41 |
| elementary_mathematics | 94.44 | 93.92 | 93.92 |
| formal_logic | 61.11 | 53.17 | 61.11 |
| global_facts | 62.00 | 64.00 | 63.00 |
| high_school_biology | 88.71 | 91.29 | 90.32 |
| high_school_chemistry | 78.33 | 75.37 | 77.34 |
| high_school_computer_science | 87.00 | 87.00 | 88.00 |
| high_school_european_history | 80.00 | 71.52 | 81.21 |
| high_school_geography | 88.89 | 88.89 | 88.38 |
| high_school_government_and_politics | 95.34 | 93.78 | 94.82 |
| high_school_macroeconomics | 84.10 | 82.56 | 86.15 |
| high_school_mathematics | 74.07 | 75.93 | 74.07 |
| high_school_microeconomics | 89.92 | 89.92 | 89.08 |
| high_school_physics | 66.89 | 73.51 | 69.54 |
| high_school_psychology | 91.56 | 91.01 | 91.74 |
| high_school_statistics | 78.24 | 77.78 | 78.70 |
| high_school_us_history | 85.78 | 77.94 | 85.78 |
| high_school_world_history | 85.23 | 75.95 | 83.12 |
| human_aging | 74.44 | 73.54 | 71.75 |
| human_sexuality | 83.21 | 81.68 | 79.39 |
| international_law | 85.12 | 85.95 | 81.82 |
| jurisprudence | 85.19 | 84.26 | 85.19 |
| logical_fallacies | 84.66 | 85.89 | 85.89 |
| machine_learning | 65.18 | 66.96 | 66.07 |
| management | 85.44 | 82.52 | 83.50 |
| marketing | 89.32 | 87.18 | 88.46 |
| medical_genetics | 86.00 | 90.00 | 87.00 |
| miscellaneous | 91.57 | 91.95 | 91.19 |
| moral_disputes | 78.03 | 75.72 | 78.32 |
| moral_scenarios | 74.53 | 75.53 | 73.85 |
| nutrition | 82.68 | 81.70 | 83.01 |
| philosophy | 75.88 | 76.21 | 77.17 |
| prehistory | 84.57 | 84.88 | 83.95 |
| professional_accounting | 75.89 | 74.11 | 75.18 |
| professional_law | 56.98 | 50.72 | 57.43 |
| professional_medicine | 77.94 | 63.60 | 79.04 |
| professional_psychology | 80.07 | 78.27 | 79.08 |
| public_relations | 68.18 | 67.27 | 66.36 |
| security_studies | 73.06 | 73.47 | 73.47 |
| sociology | 87.06 | 85.57 | 86.07 |
| us_foreign_policy | 85.00 | 85.00 | 84.00 |
| virology | 52.41 | 53.61 | 53.01 |
| world_religions | 84.21 | 83.63 | 86.55 |

Table 12: Comparison between Top-$k$/Top-$p$ and reflection-window decoding on MMLU with Llama3.1-8B

| Subject | Top-$k$/Top-$p$ | Reflection-Window |
|---|---|---|
| abstract_algebra | 43.00 | 50.00 |
| anatomy | 70.37 | 68.15 |
| astronomy | 71.05 | 75.00 |
| business_ethics | 69.00 | 71.00 |
| clinical_knowledge | 75.09 | 74.72 |
| college_biology | 80.56 | 81.94 |
| college_chemistry | 54.00 | 50.00 |
| college_computer_science | 58.00 | 47.00 |
| college_mathematics | 38.00 | 45.00 |
| college_medicine | 68.79 | 70.52 |
| college_physics | 49.02 | 65.69 |
| computer_security | 69.00 | 73.00 |
| conceptual_physics | 64.68 | 66.81 |
| econometrics | 47.37 | 48.25 |
| electrical_engineering | 57.24 | 60.00 |
| elementary_mathematics | 88.89 | 85.71 |
| formal_logic | 53.17 | 46.03 |
| global_facts | 48.00 | 41.00 |
| high_school_biology | 78.71 | 79.03 |
| high_school_chemistry | 61.08 | 62.07 |
| high_school_computer_science | 71.00 | 72.00 |
| high_school_european_history | 72.73 | 73.94 |
| high_school_geography | 78.79 | 80.81 |
| high_school_government_and_politics | 84.97 | 83.42 |
| high_school_macroeconomics | 68.97 | 73.33 |
| high_school_mathematics | 60.74 | 62.59 |
| high_school_microeconomics | 70.17 | 69.33 |
| high_school_physics | 53.64 | 53.64 |
| high_school_psychology | 85.50 | 85.50 |
| high_school_statistics | 63.89 | 65.74 |
| high_school_us_history | 77.94 | 78.92 |
| high_school_world_history | 75.11 | 79.32 |
| human_aging | 69.51 | 64.13 |
| human_sexuality | 53.44 | 54.96 |
| international_law | 78.51 | 74.38 |
| jurisprudence | 73.15 | 76.85 |
| logical_fallacies | 74.85 | 79.14 |
| machine_learning | 49.11 | 53.57 |
| management | 80.58 | 82.52 |
| marketing | 86.32 | 85.47 |
| medical_genetics | 75.00 | 76.00 |
| miscellaneous | 86.21 | 86.21 |
| moral_disputes | 63.29 | 59.25 |
| moral_scenarios | 44.47 | 49.72 |
| nutrition | 72.22 | 72.88 |
| philosophy | 69.13 | 68.81 |
| prehistory | 71.30 | 70.06 |
| professional_accounting | 47.87 | 53.19 |
| professional_law | 47.72 | 48.24 |
| professional_medicine | 76.47 | 77.94 |
| professional_psychology | 68.14 | 68.30 |
| public_relations | 69.09 | 63.64 |
| security_studies | 64.49 | 61.22 |
| sociology | 77.61 | 78.61 |
| us_foreign_policy | 81.00 | 81.00 |
| virology | 48.80 | 51.20 |
| world_religions | 78.36 | 78.36 |

Table 13: Comparison between Top-$k$/Top-$p$ and reflection-window decoding on MMLU with Mistral-Nemo

| Subject | Top-$k$/Top-$p$ | Reflection-Window |
|---|---|---|
| abstract_algebra | 34.00 | 41.00 |
| anatomy | 60.74 | 59.26 |
| astronomy | 67.76 | 66.45 |
| business_ethics | 61.00 | 56.00 |
| clinical_knowledge | 69.81 | 70.94 |
| college_biology | 74.31 | 77.08 |
| college_chemistry | 41.00 | 48.00 |
| college_computer_science | 56.00 | 54.00 |
| college_mathematics | 45.00 | 39.00 |
| college_medicine | 58.96 | 67.63 |
| college_physics | 51.96 | 52.94 |
| computer_security | 64.00 | 69.00 |
| conceptual_physics | 63.40 | 61.70 |
| econometrics | 51.75 | 57.89 |
| electrical_engineering | 55.17 | 53.10 |
| elementary_mathematics | 73.81 | 74.87 |
| formal_logic | 42.06 | 42.86 |
| global_facts | 45.00 | 41.00 |
| high_school_biology | 76.77 | 75.16 |
| high_school_chemistry | 51.72 | 57.14 |
| high_school_computer_science | 70.00 | 73.00 |
| high_school_european_history | 66.67 | 66.06 |
| high_school_geography | 75.76 | 70.20 |
| high_school_government_and_politics | 80.83 | 84.46 |
| high_school_macroeconomics | 66.92 | 69.49 |
| high_school_mathematics | 59.63 | 60.00 |
| high_school_microeconomics | 65.97 | 69.33 |
| high_school_physics | 43.71 | 45.70 |
| high_school_psychology | 78.72 | 77.80 |
| high_school_statistics | 58.80 | 62.50 |
| high_school_us_history | 64.71 | 73.04 |
| high_school_world_history | 70.46 | 73.42 |
| human_aging | 61.43 | 58.30 |
| human_sexuality | 61.83 | 64.89 |
| international_law | 68.60 | 66.94 |
| jurisprudence | 67.59 | 64.81 |
| logical_fallacies | 69.94 | 65.64 |
| machine_learning | 49.11 | 51.79 |
| management | 69.90 | 69.90 |
| marketing | 76.50 | 71.79 |
| medical_genetics | 77.00 | 72.00 |
| miscellaneous | 79.69 | 79.57 |
| moral_disputes | 58.38 | 60.12 |
| moral_scenarios | 26.15 | 27.49 |
| nutrition | 67.32 | 65.03 |
| philosophy | 61.41 | 67.85 |
| prehistory | 61.11 | 61.11 |
| professional_accounting | 46.10 | 47.16 |
| professional_law | 44.39 | 44.46 |
| professional_medicine | 58.46 | 58.46 |
| professional_psychology | 61.93 | 65.69 |
| public_relations | 65.45 | 59.09 |
| security_studies | 52.65 | 55.92 |
| sociology | 68.66 | 72.64 |
| us_foreign_policy | 75.00 | 67.00 |
| virology | 37.95 | 42.17 |
| world_religions | 71.35 | 76.02 |

Table 14: Comparison between Top-$k$/Top-$p$ and reflection-window decoding on MMLU with Phi3-Medium

| Subject | Top-$k$/Top-$p$ | Reflection-Window |
|---|---|---|
| abstract_algebra | 52.00 | 62.00 |
| anatomy | 70.37 | 72.59 |
| astronomy | 82.89 | 82.24 |
| business_ethics | 72.00 | 77.00 |
| clinical_knowledge | 78.87 | 79.25 |
| college_biology | 82.64 | 81.25 |
| college_chemistry | 51.00 | 54.00 |
| college_computer_science | 61.00 | 63.00 |
| college_mathematics | 50.00 | 48.00 |
| college_medicine | 69.94 | 72.25 |
| college_physics | 71.57 | 63.73 |
| computer_security | 74.00 | 74.00 |
| conceptual_physics | 75.74 | 74.47 |
| econometrics | 57.89 | 59.65 |
| electrical_engineering | 60.00 | 63.45 |
| elementary_mathematics | 85.98 | 85.71 |
| formal_logic | 56.35 | 57.14 |
| global_facts | 45.00 | 46.00 |
| high_school_biology | 84.52 | 86.45 |
| high_school_chemistry | 74.38 | 73.40 |
| high_school_computer_science | 81.00 | 83.00 |
| high_school_european_history | 73.33 | 75.76 |
| high_school_geography | 81.82 | 82.83 |
| high_school_government_and_politics | 88.60 | 91.19 |
| high_school_macroeconomics | 80.00 | 79.74 |
| high_school_mathematics | 60.37 | 63.70 |
| high_school_microeconomics | 84.03 | 87.39 |
| high_school_physics | 58.28 | 70.20 |
| high_school_psychology | 85.50 | 85.50 |
| high_school_statistics | 70.83 | 72.69 |
| high_school_us_history | 82.84 | 78.92 |
| high_school_world_history | 81.01 | 81.86 |
| human_aging | 71.30 | 68.16 |
| human_sexuality | 73.28 | 77.10 |
| international_law | 77.69 | 83.47 |
| jurisprudence | 83.33 | 75.00 |
| logical_fallacies | 79.14 | 82.82 |
| machine_learning | 54.46 | 64.29 |
| management | 76.70 | 80.58 |
| marketing | 83.76 | 81.62 |
| medical_genetics | 85.00 | 87.00 |
| miscellaneous | 85.31 | 84.55 |
| moral_disputes | 71.97 | 67.92 |
| moral_scenarios | 61.34 | 63.24 |
| nutrition | 75.82 | 78.10 |
| philosophy | 74.92 | 71.06 |
| prehistory | 76.85 | 78.09 |
| professional_accounting | 68.09 | 66.67 |
| professional_law | 53.13 | 50.00 |
| professional_medicine | 69.12 | 67.65 |
| professional_psychology | 76.31 | 76.96 |
| public_relations | 64.55 | 69.09 |
| security_studies | 71.43 | 66.12 |
| sociology | 83.08 | 84.58 |
| us_foreign_policy | 81.00 | 84.00 |
| virology | 49.40 | 49.40 |
| world_religions | 77.78 | 74.85 |

Table 15: Comparison between Top-$k$/Top-$p$ and reflection-window decoding on MMLU with Qwen2.5-14B

| Subject | Top-$k$/Top-$p$ | Reflection-Window |
|---|---|---|
| abstract_algebra | 78.00 | 81.00 |
| anatomy | 71.85 | 76.30 |
| astronomy | 88.16 | 91.45 |
| business_ethics | 76.00 | 75.00 |
| clinical_knowledge | 81.51 | 82.64 |
| college_biology | 90.97 | 86.11 |
| college_chemistry | 64.00 | 60.00 |
| college_computer_science | 73.00 | 73.00 |
| college_mathematics | 72.00 | 80.95 |
| college_medicine | 73.41 | 90.91 |
| college_physics | 82.35 | 84.31 |
| computer_security | 82.00 | 84.00 |
| conceptual_physics | 82.55 | 84.68 |
| econometrics | 61.40 | 65.79 |
| electrical_engineering | 73.10 | 73.79 |
| elementary_mathematics | 95.24 | 94.97 |
| formal_logic | 67.46 | 61.11 |
| global_facts | 51.00 | 52.00 |
| high_school_biology | 88.39 | 90.97 |
| high_school_chemistry | 80.30 | 75.37 |
| high_school_computer_science | 91.00 | 89.00 |
| high_school_european_history | 76.97 | 81.82 |
| high_school_geography | 89.39 | 88.89 |
| high_school_government_and_politics | 91.19 | 92.75 |
| high_school_macroeconomics | 85.64 | 86.41 |
| high_school_mathematics | 84.44 | 84.27 |
| high_school_microeconomics | 89.08 | 88.60 |
| high_school_physics | 82.78 | 79.47 |
| high_school_psychology | 90.09 | 91.38 |
| high_school_statistics | 83.33 | 81.48 |
| high_school_us_history | 87.25 | 88.73 |
| high_school_world_history | 88.61 | 86.92 |
| human_aging | 73.54 | 73.99 |
| human_sexuality | 74.05 | 81.68 |
| international_law | 80.17 | 77.69 |
| jurisprudence | 78.70 | 84.26 |
| logical_fallacies | 83.44 | 85.28 |
| machine_learning | 72.32 | 74.11 |
| management | 82.52 | 85.44 |
| marketing | 88.46 | 90.60 |
| medical_genetics | 89.00 | 91.00 |
| miscellaneous | 89.40 | 91.42 |
| moral_disputes | 71.97 | 72.54 |
| moral_scenarios | 69.61 | 68.49 |
| nutrition | 82.35 | 81.70 |
| philosophy | 78.78 | 80.06 |
| prehistory | 86.11 | 83.02 |
| professional_accounting | 73.76 | 71.99 |
| professional_law | 54.04 | 56.00 |
| professional_medicine | 81.62 | 86.40 |
| professional_psychology | 77.94 | 79.08 |
| public_relations | 71.82 | 71.82 |
| security_studies | 71.43 | 73.06 |
| sociology | 85.07 | 88.06 |
| us_foreign_policy | 87.00 | 84.00 |
| virology | 50.00 | 51.81 |
| world_religions | 84.21 | 84.80 |

