# OpenReview forum: "Reflection-Window Decoding: Text Generation with Selective Refinement"
_ICML.cc/2025/Conference — ICML 2025 poster_

### Official Review · Reviewer_ZxMG · 2025-02-24

**Overall Recommendation:** 3

**Summary:**

This paper introduces a built-in mechanism for refinement and correction during LLM generation. The authors provide theoretical analysis characterizing the sub-optimality of purely autoregressive decoding and propose a "reflection-window" decoding approach that allows for selective refinement during generation. The method shows improved empirical results on multiple benchmarks.

**Claims And Evidence:**

The potential deviation of auto-regressive generation is supported by theory and demonstrative examples.

**Essential References Not Discussed:**

No

**Experimental Designs Or Analyses:**

Some experimental designs are not convincing:

1. In Figure 5, the authors compare the win rate between Beam Search and Reflection-Window (Greedy). However, since Reflection-Window (Greedy) primarily relies on greedy decoding with occasional (3.5%–5.5%) use of Beam Search, it is essential to include pure greedy decoding as a baseline for comparison. I strongly encourage the authors to add this comparison.

2. In Table 2, the authors compare the performance between Top-K/Top-P and Reflection-Window. The improved performance of Reflection-Window primarily highlights the effectiveness of Beam Search over Top-K/Top-P in this task. I strongly encourage the authors to include a pure Beam Search baseline for comparison.

**Methods And Evaluation Criteria:**

The proposed methods exhibit some deviations from the dependence structure illustrated in Figure 1(a).

**Other Comments Or Suggestions:**

None

**Other Strengths And Weaknesses:**

**Stengths**
Clear motivations supported by theoretical analysis and demonstrative examples.

**Weaknesses**
I am mostly concerned with the experimental parts. Please refer to Experimental Designs Or Analyses. Besides, in Table 1, Reflection-Window (Greedy) shows no improvement over purely greedy decoding.

**Questions For Authors:**

A more rigorous experimental evaluation and positive results would change my score.

**Relation To Broader Scientific Literature:**

None

**Theoretical Claims:**

The proofs are sound.

---

> ### Author Rebuttal · Authors · 2025-04-01
>
> # Response to Reviewer ZxMG
>
> Thank reviewer for the comments and questions, as well as the time devoted! Please kindly notice that there might be some **potential misunderstandings**. Below please see our point-by-point responses:
>
> ---
>
> ### **C1:** "In Figure 5, the authors compare the win rate between Beam Search and Reflection-Window (Greedy). However, since Reflection-Window (Greedy) primarily relies on greedy decoding with occasional (3.5%–5.5%) use of Beam Search, it is essential to include pure greedy decoding as a baseline for comparison. [The reviewer] strongly encourage the authors to add this comparison."
>
> **A1:** Thanks for considering our experimental results. There might be some **potential misinterpretation**.
>
> In Figure 5, we present the win rate of beam search _against_ greedy decoding, and that of our approach _against_ greedy decoding. In other words, the win/lose is not calculated between beam search and our approach, but compared against greedy decoding, respectively. We think the brief caption (due to space limit) might be the reason behind this misinterpretation, and we have modified it into "Figure 5: Comparison of win rates of beam search and our reflection-window decoding (both against greedy decoding) on MT-Bench across categories."
>
> ---
>
> ### **C2:** "In Table 2, the authors compare the performance between Top-K/Top-P and Reflection-Window. The improved performance of Reflection-Window primarily highlights the effectiveness of Beam Search over Top-K/Top-P in this task. [The reviewer] strongly encourage the authors to include a pure Beam Search baseline for comparison."
>
> **A2:** Thanks for the comment. There might be some **potential overlook** in our specified setting.
>
> In Table 2, the "regular decoding" in our approach is Top-$k$/Top-$p$, and the difference between our approach and the vanilla Top-$k$/Top-$p$ baseline includes both the **sliding-window reflection** and the **selective refinement** mechanisms. Describing Table 2 as primarily highlighting the effectiveness of beam search might oversimplify the results, potentially overlooking the substantial improvements in computational efficiency and performance introduced by these additional mechanisms.
>
> Regarding the pure beam search baseline, we included the evaluation results in Table 11, and the pointer was provided on line 344 (Right).

---

### Official Review · Reviewer_Fghj · 2025-03-14

**Overall Recommendation:** 2

**Summary:**

This proposes Reflection-Window decoding, an attempt at addressing the limitations of autoregressive text generation in large language models (LLMs), which lack built-in mechanisms for refining or correcting generated content. The authors analyze how sequential token-by-token optimization can deviate from a globally optimal response and propose an alternative approach called Reflection-Window Decoding. This method introduces a sliding reflection window and a pausing criterion, enabling selective refinement of text as it is generated. By balancing efficiency and optimality, the proposed framework improves text generation quality, outperforming traditional autoregressive decoding while maintaining efficiency comparable to beam search. Extensive empirical evaluations validate the effectiveness of this approach in mitigating suboptimalities inherent in existing decoding strategies.

**Claims And Evidence:**

- Decoding towards step-wise MAP of the autoregressive model is suboptimal - Non-trivially correct but this is well-known by the community. People still following the current practices just because this is a rather good enough approximation.
- Saliency-based reflection - This is a rather interesting point, but I don't think its deeper insightly is sufficiently discussed and/or proven.
- Compatibility and versatility of the proposed method - This is mostly right to me.
- Improved efficiency with less sacrificing of quality - True but the empirical results kinda weak and the significance is also less consistent given different base models.

**Essential References Not Discussed:**

I am not an expert in this domain, to the best of my knowledge, apart from the aforementioned direct approaches, there is not any essential references that I know omitted by the authors.

**Experimental Designs Or Analyses:**

The experiment is well formulated and correctly conducted.

**Methods And Evaluation Criteria:**

The evalutation is standard and correctly conducted.

**Other Comments Or Suggestions:**

Using the fluctuation of likelihood to determine potential reflection is an interesting point. I wonder if the authors could combine this with contrastive methods (e.g. before computation of saliency first baselining the likelihood by a smaller proxy model) to achieve more compelling results.

**Other Strengths And Weaknesses:**

This paper, while not very empirically strong, provides an interesting point of view for the community to rethink between the choice of "computation through tokens" and "computation through logits". In this particular case, it is "reflection as tokens" and "reflection as logits' (change in values)". This might reveal a deeper, unified story about how reflection in LLM works that could facilitate future research.

**Questions For Authors:**

What's your understanding about the scalability of your proposed method, i.e. to what extent, it could enable the models to do complex tasks that it couldn't without the approach?

**Relation To Broader Scientific Literature:**

There are a few more recent works/commercial practices (such as OpenAI O1/3 and DeepSeek R1) that uses a combination of reflection-aware chain-of-thought SFT and RL to achieve a similar computational goal of the proposed method. While I agree with the authors that this proposed method has the merit of minimising the additional overhead in achieving better results, unfortunately (yes, yet another bitter lesson), it is less extensive and scalable than those rather direct methods to introduce the reflection mechanism through non-architectural/algorithmic ways.

**Theoretical Claims:**

There is but one strictly verified theorem A.1 that is non-trivially correct yet kinda well-known and thus not surprising to the community. The other major theorem 3.6, as is admitted by the authors, provides a bound that is too weak to be practically useful. While the general framework of the proposed method is of great potential and very interesting, the current progress of the algorithm is kind of disappointing and less theoretically supported than it might seem.

---

> ### Author Rebuttal · Authors · 2025-04-01
>
> # Response to Reviewer Fghj
>
> We are very grateful for the thoughtful comments, as well as the time and effort devoted! Below please see our point-by-point responses:
>
> ---
>
> ### **C1:** "There are a few more recent works/commercial practices (such as OpenAI O1/3 and DeepSeek R1) that uses a combination of reflection-aware chain-of-thought SFT and RL to achieve a similar computational goal of the proposed method."
>
> **A1:** Thanks for the comment. We completely agree that recent works/commercial practices have explored various methods to enhance generated content (Sections 1 and 2). However, the fundamental limitation of autoregressive decoding itself remains under-explored. This gap represents a distinct perspective, different from high-level model behaviors or inference efficiency.
>
> Furthermore, as you kindly pointed out, our method is versatile and provides practitioners additional flexibilities to incorporate different strategies (without the need to retrain or finetune).
>
> ---
>
> ### **C2:** "[The paper] provides an interesting point of view for the community to rethink between the choice of 'computation through tokens' and 'computation through logits.' In this particular case, it is 'reflection as tokens' and 'reflection as logits' (change in values). This might reveal a deeper, unified story about how reflection in LLM works that could facilitate future research."
>
> **A2:** Thanks for sharing the insight and the perspective! In light of your comment, we would very much like to include such discussion in the paper, and we sincerely hope that our work can facilitate future research. It would be greatly appreciated if you can kindly share pointers to where these terms were discussed.
>
> ---
>
> ### **C3:** "[The reviewer] wonder if the authors could combine this with contrastive methods (e.g. before computation of saliency first baselining the likelihood by a smaller proxy model) to achieve more compelling results."
>
> **A3:** Thanks for the thoughtful question. If we understood the comment correctly, by "contrastive methods" we are discussing the compatibility of our framework with contrastive decoding (CD), e.g., Li et al. (2023), O'Brien and Lewis (2023). CD utilizes a search-based decoding approach that contrasts LMs of different scales, for instance, an expert (larger LM) and an amateur (smaller LM). In principle, our approach is versatile and compatible with contrastive methods, and can be applied in various ways.
>
> For instance, one can apply our reflection-window decoding on expert and amateur LLMs in parallel, and then continue with CD's method of factoring out undesired behaviors of smaller LMs while retaining good behaviors of larger ones. Alternatively, one can also apply CD first, and then design appropriate pausing criterion to incorporate our selective refinement framework. We leave these as interesting directions for future works.
>
> Please kindly let us know if we accidentally misunderstood your comment.
>
> ---
>
> ### **Q4:** "What's your understanding about the scalability of your proposed method, i.e. to what extent, it could enable the models to do complex tasks that it couldn't without the approach?"
>
> **A4:** Thanks for the thoughtful question and for trying to go further. As the primary goal of this paper is to address the pitfall of the purely autoregressive way of decoding, we do not claim that our approach can enable high-level behaviors (e.g., complex new tasks) that are otherwise unattainable. However, as you kindly pointed out in **C2**, we sincerely hope our perspective can provide a distinct point of view for community to rethink about related issues, and facilitate future research.
>
> ---
>
> ### References
>
> Li, Xiang Lisa, Ari Holtzman, Daniel Fried, Percy Liang, Jason Eisner, Tatsunori Hashimoto, Luke Zettlemoyer, and Mike Lewis. "Contrastive Decoding: Open-ended Text Generation as Optimization." _The 61st Annual Meeting Of The Association For Computational Linguistics_. 2023.
>
> O'Brien, Sean, and Mike Lewis. "Contrastive decoding improves reasoning in large language models." arXiv preprint arXiv:2309.09117 (2023).

---

### Official Review · Reviewer_Lq2Q · 2025-03-17

**Overall Recommendation:** 3

**Summary:**

The paper makes the observation that, given an autoregressive language model $p_{LM}$, the token sequences $\mathbf{\hat{x}}$ generated via greedy decoding does not always correspond to the MAP state $\mathbf{x}^* = argmax_{\mathbf{x}} p\_{LM}(\mathbf{x})$. Theoretical analysis shows that under mild assumptions, at step $L < T$ of the autoregressive generation, if $p\_{LM}(\mathbf{\hat{x}}\_{\leq L})$ falls below $p\_{LM} ({\mathbf{x}^*}\_{\leq L})$,  then (1) the next-token probability $p\_{LM}(\mathbf{\hat{x}}\_L | \mathbf{\hat{x}}\_{< L})$ falls under some threshold and (2) there exists some $K < L$ s.t. $\mathbf{\hat{x}}\_{\leq K}$ differ from $\mathbf{x}^*\_{\leq K}$ (if I interpret Theorem 3.6 correctly). To overcome this sub-optimality of greedy decoding, as an alternative to beam search, the authors propose an approach where: (1) we can pause autoregressive generation depending on certain criteria (e.g. the entropy of the next-token distribution as suggested by Theorem 3.6) and (2) then regenerate the last $d$ tokens via, e.g. beam search, to maximize the probability. Empirical evaluations on benchmarks such as MMLU and MT-bench show that the proposed approach consistently outperforms greedy decoding and beam search, as well as top-p and top-k sampling.

**Claims And Evidence:**

Theorem 3.6 needs further clarification. In particular, the probabilistic semantics of the threshold $\epsilon\_{L}$ is unclear, i.e., what does the ratio mean or how does it correlate with the discrepancy between the sequence obtained via greedy decoding and the MAP state? Without such explanation, it's kind of hard to interpret why we want to regenerate the last $d$ tokens when $\epsilon\_{L}$ is "small". Further, is it possible to analyze when the proposed approach is guaranteed to generate sequences with probability higher than beam search?

**Essential References Not Discussed:**

Perhaps discuss the relationship between this work and *Shih, Andy, Dorsa Sadigh, and Stefano Ermon. "Long horizon temperature scaling." International Conference on Machine Learning. PMLR, 2023.*

**Experimental Designs Or Analyses:**

Yes.

**Methods And Evaluation Criteria:**

Though I agree that the ultimate goal of new decoding algorithms is to improve model performance on downstream applications, to support the argument made by this paper, it would be more helpful if the authors can also directly compare the probability of the sequences generated by beam search, greedy decoding and the proposed approach. For example, Table 1 suggests that beam search performs worse than greedy decoding, which is kind of counter-intuitive as the sequences generated from beam search should always have higher probability than that from greedy decoding (correct me if I'm wrong). This is probably suggesting that higher probability does not always imply better accuracy/fluency etc. However, this is not a huge problem: as long as the proposed approach can effectively boost the probability of the generations, the main argument made by this paper should already be well-supported.

**Other Comments Or Suggestions:**

The statement of Theorem 3.6, as well as the whole Sec. 3, should be improved. The notations used are overly complicated. For example, I don't see why $\mathbf{v}$ and $T$ need to be carried everywhere throughout the section: $T$ can be assumed to be some constant and omitted and $v$ seem to serve as the same functionality as $w$ and $X_t$. More specifically, Defn 3.1 is really just defining the sequence obtained from greedy decoding and Defn 3.2 is just defining the sequence that maximizes the joint probability. I don't see why something like $\mathbf{\hat{x}}$ and $\mathbf{x}^*$ are not sufficient.

**Other Strengths And Weaknesses:**

See above.

**Questions For Authors:**

See above.

**Relation To Broader Scientific Literature:**

The contributions of this paper is very relevant to language generation in general, the proposed approach can be viewed as a generalization of beam search, which has been commonly used for LLM decoding.

**Theoretical Claims:**

Yes.

---

> ### Author Rebuttal · Authors · 2025-04-01
>
> # Response to Reviewer Lq2Q
>
> Thanks for the thoughtful and detailed comments, as well as the time and effort devoted! Below please see our responses to specific comments and questions:
>
> ---
>
> ### **Q1:** "[Theorem 3.6] what does the ratio $\epsilon_L$ mean or how does it correlate with the discrepancy between the sequence obtained via greedy decoding and the MAP state? why we want to regenerate the last $d$ tokens when $\epsilon_L$ is 'small'"
>
> **A1:** Thanks for asking about the $\epsilon_L$ term in our theoretical result. The denominator is the ground-truth joint probability of the length-$(L-1)$ stepwise-optimal response (obtained by greedy decoding), and the numerator is that of the length-$L$ globally-optimal response (obtained by MAP). Theorem 3.6 roughly states that if the model is very uncertain when trying to generate the $L$-th token, then there is an error in the generation history at the $K$-th token, and that $K < L$. Therefore, we need to get back to the token $K$ to start the revision.
>
> In practice, we look back $d$ tokens and regenerate them. We also provide additional discussions on the choice of $d$ in Section 5.4 and Appendix B.1.
>
> ---
>
> ### **Q2:** "Is it possible to analyze when the proposed approach is guaranteed to generate sequences with probability higher than beam search?"
>
> **A2:** Thanks for the thoughtful question and for trying to go further. If there is no limit on computation and storage, unconstrained-beam-width beam search could yield the globally optimal output (through actual MAP), i.e., guaranteed to be of the highest probability and to outperform (or at least no worse than) any other approach (including the proposed approach).
>
> With a fixed-beam-width beam search, the theoretical characterization of the generated length-$L$ sequence is highly nontrivial, since the frontier depends on the pruning at all previous steps. Please feel free to let us know if you would like suggest a way to perform such theoretical analysis.
>
> ---
>
> ### **C3:** "Table 1 [...] the sequences generated from beam search should always have higher probability than that from greedy decoding (correct [the reviewer] if [they were] wrong)"
>
> **A3:** Thanks for carefully considering our results. The metric in Table 1 is accuracy instead of joint probability.
>
> In benchmark evaluations, it is difficult to precisely control the output length of different decoding methods. Directly setting a hard cutoff of token numbers may yield incomplete/insensible responses. Therefore, in Table 1 we present accuracies on MMLU (consistent with previous works).
>
> ---
>
> ### **C4:** "Discuss the relation with Shih et al. (2023)"
>
> **A4:** Thanks for providing the pointer to a related work!
>
> Shih et al. (2023) proposed _Long Horizon Temperature Scaling_ (LHTS), which samples from temperature-scaled joint distributions, to address the myopic temperature scaling in autoregressive models. LHTS optimizes for the long horizon likelihood of samples, and can enable a model to generate with a controllable long horizon temperature parameter through finetuning. In comparison, our work aims to address the pitfall of the purely autoregressive generation itself, and our approach is versatile and compatible with LHTS (Shih et al., 2023).
>
> In light of your comment, we will incorporate the above discussion in the revised paper. Thanks again for providing the pointer.
>
> ---
>
> ### **C5:** "Sec. 3 the notations used are overly complicated. For example, [the reviewer] don't see why $\mathbf{v}$ and $T$ need to be carried everywhere [..., the reviewer] don't see why something like $\widehat{\mathbf{x}}$ and $\mathbf{x}^*$ are not sufficient"
>
> **A5:** Thanks for the comment on notation. When presenting the theoretical result, we explicitly keep length indices since lengths play an important role when evaluating the joint probability of a response. The longer the length, the lower the probability tends to be, and this is the case for both $\widehat{\mathbf{x}}$'s and $\mathbf{x}^*$'s.
>
> For instance (we provided this example in lines 165--177),  if we were to use 10 words to distinguish between joint and conditional densities, one might say "joint density combines all variables; conditional adjusts for known variables." However, if we can use 15 words, one might say "joint density reflects combined probabilities of all variables; conditional density adjusts probabilities given known variables." A fair comparison between $\widehat{\mathbf{x}}$ and $\mathbf{x}^*$ should be length specific. Therefore, we think $\widehat{\mathbf{x}}$ and $\mathbf{x}^*$ together with length indices help make this subtlety more transparent.
>
> ---
>
> ### Reference
>
> Shih, Andy, Dorsa Sadigh, and Stefano Ermon. "Long horizon temperature scaling." _International Conference on Machine Learning_. PMLR, 2023.

---

### Official Review · Reviewer_KVvm · 2025-03-18

**Overall Recommendation:** 3

**Summary:**

The authors describe, theoretically and empirically, how greedy sampling is suboptimal for generating the sentence with maximum likelihood.
They then propose an alternative algorithm, which pauses the generation when a specific criterion is triggered, and regenerates a small portion of the text.
The proposed method is evaluated on several models and datasets, which shows its superiority over standard decoding strategies.

**Claims And Evidence:**

The main claims of the paper are regarding the suboptimality of greedy decoding. The authors do a good job are convincing the reader that this is indeed the case.
The necessity to change the generation algorithm in order to generate a globally optimal sentence seems natural.

**Essential References Not Discussed:**

None to the best of my knowledge.

**Experimental Designs Or Analyses:**

Overall the authors evaluate their method on a few different models and datasets. It would still be more convincing to see more diverse evaluation, especially when the results seem to be noisy, with sometimes marginal gains.

**Methods And Evaluation Criteria:**

The main algorithm is interesting and flexible.
The choice of entropy as a pausing criterion is well motivated. However the criterion requires the entropy of all past $d$ tokens to be above the threshold, which seems less natural to me. The criterion would not trigger if the LLM is very uncertain about only one token. Is that a desirable behavior? It may be interesting to evaluate and compare difference choices for this pausing criterion.

How come the proposed algorithm performs better than beam search, when it is supposed to be a cheaper approximation? It seems there are some things going on that are not aligned with the theoretical explanation and motivations.

**Other Comments Or Suggestions:**

Since it seems the end algorithm is a faster approximation of beam search, it would be great to measure the actual speedup.

**Other Strengths And Weaknesses:**

The writing is not always very clear.

**Questions For Authors:**

What model is used in the synthetic setting? The setup for these experiments needs to be more detailed.

**Relation To Broader Scientific Literature:**

The paper addresses the important problem of the suboptimality of greedy decoding. It describes well the context, the motivation, and the related literature.

**Theoretical Claims:**

The paper starts with a theoretical analysis to characterize the behavior of greedy decoding compared. The claims seem mathematically founded and are quite intuitive.
However, assumption 3.3 about the LLM being an oracle (i.e. computes the exact conditional probabilities) seems farfetched. The paper could benefit from more discussion about this, for instance by comparing results for different sizes of models (smaller models are even less likely to be oracles). An indicator that this assumption may be wrong is when we notice that beam seems to perform noticeably worse than greedy decoding.

---

> ### Author Rebuttal · Authors · 2025-04-01
>
> # Response to Reviewer KVvm
>
> We are very grateful for the insightful questions and constructive comments! Below please see our point-by-point response:
>
> ---
>
> ### **Q1:** "the criterion requires the entropy of all past $d$ tokens to be above the threshold [..., but] would not trigger if the LLM is very uncertain about only one token. Is that a desirable behavior?"
>
> **A1:** Thanks for carefully considering the pausing criterion. Yes, this is indeed a desirable behavior. Uncertainty can occur when the model does not know how to proceed (due to a previous error) or when there are multiple valid ways to proceed. Therefore, we aim to capture _the trend of_ increasing uncertainty, reducing the false-positive triggerings while maintaining a low computational overhead.
>
> ---
>
> ### **C2:** "It may be interesting to evaluate and compare difference choices for this pausing criterion."
>
> **A2:** We totally agree, and this is exactly why we present Table 4 (varying entropy threshold $\sigma$) and Table 5 (varying window size $d$) in Section 5.4, and provide additional discussions and analyses (due to space limit) in Appendix B.1 - B.3, on window size, entropy threshold, and modification rate, respectively.
>
> ---
>
> ### **Q3:** "How come the proposed algorithm performs better than beam search, when it is supposed to be a cheaper approximation?"
>
> **A3:** Thanks for the thoughtful question. If there is no limit on the computation and storage, unconstrained-beam-width beam search will yield the globally optimal output through brute force. However in practice, since maintaining a full frontier quickly becomes intractable and a fixed beam width is often introduced as a hyperparameter.
>
> Our algorithm can perform better since it tackles errors as the generation goes on, while vanilla beam search needs a larger beam width to be able to enclose all possible sequences that can be generated by our approach.
>
> ---
>
> ### **C4:** "Assumption 3.3 about the LLM being an oracle (i.e. computes the exact conditional probabilities) seems farfetched"
>
> **A4:** Thanks for carefully thinking about our theoretical results. There might be potential misunderstandings, please allow us to clarify in twofold:
>
> (1) Together with Assumption 3.3, our theoretical results indicate that, **even if** with an oracle LLM, there is still suboptimality in the purely autogressive way of decoding. In other words, even if the LLM itself perfectly decomposes the (conditional) probabilities (which, as you pointed out, is a farfetched benefit to assume in practice), there is still no guarantee in obtaining the globally optimal sequence with purely autoregressive decoding.
>
> (2) The purpose of Assumption 3.3 is to facilitate clear theoretical results, and our empirical evaluation does not rely on or employ this assumption.
>
> In light of your comment, we have included above clarifications in our revised draft.
>
> ---
>
> ### **C5:** "Overall the authors evaluate their method on a few different models and datasets. It would still be more convincing to see more diverse evaluation."
>
> **A5:** Thanks for the comment. In our empirical evaluations:
>
> - for **models**, we utilize models from different families, including Llama-3.1-8B-Instruct, Phi-3-Medium-128K-Instruct, Qwen2.5-14B-Instruct, Qwen2.5-7B-Instruct, Mistral-Nemo-Instruct-2407;
> - for **benchmarks**, we consider MMLU (which include 57 diverse subjects, e.g., humanities, STEM, and social sciences, at varying difficulty levels) for evaluating reasoning performance and factual knowledge, and also MT-Bench for a fine-grained evaluation through multi-turn conversational task, including correctness, coherence, and fluency.
>
> We provide pointers to our empirical results in List of Tables in Page 11. Please just kindly let us know if you have a specific evaluation task in mind.
>
> ---
>
> ### **C6:** "Since it seems the end algorithm is a faster approximation of beam search, it would be great to measure the actual speedup."
>
> **A6:** We totally agree, and that's why in Section 5.4 paragraph "Efficiency of Reflection-Window Decoding" we provide regeneration metrics by MMLU categories (humanities, STEM, social sciences, others), and also present additional analysis in Appendix B.3.
>
> ---
>
> ### **Q7:** "What model is used in the synthetic setting? The setup for these experiments needs to be more detailed."
>
> **A7:** Thanks for asking about the detail of our synthetic setting.
>
> We use Llama-3.1-8B-Instruct. For each prompt, together with a certain length $\{0, 20, 50, 200\}$ of generation history ($0$ means only the prompt is given), we evaluate whether the joint probability of the sequence generated with greedy decoding is greater than or equal to that produced by (fixed-beam-width, set to $10$) beam search (as proxy of the global optimum). This comparison indicates the extent to which greedy decoding deviates from the globally optimal response.
>
> In light of your comment, we have included above detail in the revised draft.

---

### Decision · Program_Chairs · 2025-05-01

**Decision:**

Accept (poster)

**Comment:**

This paper studies the suboptimality of greedy decoding in autoregressive language models and proposes Reflection-Window Decoding, allowing for selective, uncertainty-triggered revision during generation. Both theory and experiments, spanning multiple models and benchmarks, support the effectiveness of the approach.

**Review Summary:**
- **Reviewer KVvm** values the theoretical foundation and empirical evaluation, raising some concerns about the pausing criterion and broader evaluation. These were addressed in the rebuttal, and the reviewer supports acceptance.
- **Reviewer Lq2Q** finds the contribution relevant and mostly well-argued, suggesting clarification for some theoretical results and asking for comparison of log-likelihoods. After rebuttal, the reviewer is satisfied and leans toward acceptance.
- **Reviewer ZxMG** initially questioned some baseline comparisons but was satisfied by the author’s clarifications and raised their score to weak accept.
- **Reviewer Fghj** appreciates the originality and potential of the work, though notes the experiments could be stronger. They remain somewhat skeptical but acknowledge the value of the new perspective.

**Author Response:**
The authors responded thoroughly, clarifying theoretical points, providing additional results, and committing to address remaining suggestions.

**Conclusion:**
With most reviewers supporting acceptance and author response effectively addressing concerns, I recommend **accept** for this submission.

---